# The spatiotemporal scaling laws of urban population dynamics

Xingye Tan [1,8], Bo Huang [1,2,3,4,8] ✉, Michael Batty [5], Weiyu Li[6], Qi Ryan Wang [7], Yulun Zhou[4] & Peng Gong [1,3]

Human mobility is becoming increasingly complex in urban environments. However, our fundamental understanding of urban population dynamics, particularly the pulsating fluctuations occurring across different locations and timescales, remains limited. Here, we use mobile device data from large cities and regions worldwide combined with a detrended fractal analysis to uncover a universal spatiotemporal scaling law that governs urban population fluctuations. This law reveals the scale invariance of these fluctuations, spanning from city centers to peripheries over both time and space. Moreover, we show that at any given location, fluctuations obey a time-based scaling law characterized by a spatially decaying exponent, which quantifies their relationship with urban structure. These interconnected discoveries culminate in a robust allometric equation that links population dynamics with urban densities, providing a powerful framework for predicting and managing the complexities of urban human activities. Collectively, this study paves the way for more effective urban planning, transportation strategies, and policies grounded in population dynamics, thereby fostering the development of resilient and sustainable cities.

Human mobility, often considered the lifeblood of a city, significantly impacts its economy, vibrancy, energy consumption, and public health[1]. This constant movement generates not only traffic patterns but also dynamic fluctuations in the population as people move in and out of various locations. Like a city's heartbeat, these fluctuations mirror the diverse human activities occurring both spatially and temporally[2]. Uncovering the patterns behind these pulsating dynamics can provide critical insights into urban functionality, growth, and resilience—all of which are essential for urban planning, environmental management, traffic prediction, and epidemic prevention[3–10].

Numerous studies have delved into the realm of human mobility, focusing on trajectories and flows[1]. Many of these works have revealed statistical regularities in scaling, such as power-law or exponential distributions, and in travel costs, including travel displacement and/or

travel time, that characterize these trajectories[11–17]. Models for estimating static flows between locations, including those based on gravitational analogies, have also been well developed. Many of these models rely on spatial associations between population movements and various indicators of attractiveness, such as population size[6,17–22]. Additionally, some studies have quantitatively demonstrated how urban spatial structures influence human travel patterns[5,10,17,23]. Furthermore, research focusing on time-variant human processes has sought to identify, categorize, and visualize temporal patterns of human activity across diverse urban settings[3,4,24]. These efforts have significantly advanced our understanding of the patterns and behaviors associated with human movement in urban environments.

However, the population fluctuations driven by human mobility, which can generally be regarded as occurring along a dimension

[1]Department of Geography, The University of Hong Kong, Hong Kong SAR, China. [2]Computational Social Science Laboratory, Faculty of Social Science, The University of Hong Kong, Hong Kong SAR, China. [3]Urban Systems Institute, The University of Hong Kong, Hong Kong SAR, China. [4]Department of Urban Planning and Design, The University of Hong Kong, Hong Kong SAR, China. [5]The Bartlett Centre for Advanced Spatial Analysis, University College London, London, UK. [6]School of Mathematical Sciences, Suzhou University of Science and Technology, Suzhou, China. [7]Department of Civil and Environmental Engineering, Northeastern University, Boston, MA, USA. [8]These authors contributed equally: Xingye Tan, Bo Huang. ✉e-mail: bohuang@hku.hk

approximately perpendicular to the plane of trajectories and flows, have never received adequate attention. This oversight may stem from the challenge of concisely and universally quantifying these dynamics over time and space, as well as from difficulties in developing a model that explains such phenomena[25]. Similar to findings in human mobility, substantial evidence demonstrates that scale invariance exists in urban systems across spatial, hierarchical, and size dimensions, suggesting the absence of characteristic scales[26–30] and the presence of fractal-like geometry[31–33]. Nevertheless, the temporal scaling behavior of urban systems remains significantly understudied. Time scaling properties have primarily been explored in biophysical and natural phenomena, such as physiological signals[34], meteorological observations[35], and hydrological processes[36]. Moreover, the relationship between temporal scaling and geographic space remains unknown. Therefore, it is vital to examine the temporal scaling behavior of population dynamics in conjunction with their spatial regularity, in order to establish more universal laws that can enrich urban science[2].

In this study, we seek to uncover the temporal and spatial scaling patterns of urban population fluctuations. Using geotagged mobile device data from five municipalities and one metropolitan area across different continents (Asia, Europe, and America; see "Methods" and Supplementary Notes 1 and 2), we apply detrended fluctuation analysis (DFA)[37] to grid-based population time series (see Fig. 1 and "Methods" for more details). Our analysis reveals that population fluctuations within urban systems exhibit spatiotemporal scale invariance, indicating a distinctive scale-free behavior in both spatial and temporal domains that is characterized by a pair of power-law functions. The scaling exponents are not constant but exhibit a spatiotemporal gradient in a logarithmic manner. Notably, this gradient pattern, where temporal scaling exponents are spatially organized at the micro level,

indicates well-defined urban spatial structures. These structures generally display an organization akin to the distance-decay layout of mobility and its influencing factors, such as population density and functional attractiveness. This is consistent with the process of urban growth spreading outward from a historic center, the standard model for the development of industrial cities and their later variants[5,17,38,39]. By associating the analysis with the conventional model of urban density, we reveal an urban allometry that elucidates the regularity of population fluctuations. These findings enhance our understanding of seemingly random yet structured human activities in relation to urban configurations, thus informing economic activities, urban planning, and public health strategies where population dynamics are highly relevant.

## Results

### Spatial and temporal scale invariance of population fluctuations

To characterize urban population fluctuations and reveal their spatiotemporal scaling patterns, we used equidistant time series grid data derived from datasets captured on and extracted from mobile devices ("Methods"). We calculated the mean population fluctuation, $F_i(s)$, of each grid cell $i$ over varying time durations $s$ (i.e., a series of given temporal scales, ranging from an hour to a day in increments of 20 min, for measuring time series statistics by considering the scaling effect). This was achieved by using DFA, where $F_i(s)$ is the square root of the averaged variance of the linearly detrended time series profile (see "Methods"). Furthermore, we defined and measured the mean population fluctuation, $F(s, r)$, at distance $r$ from the main urban center within these time intervals $s$. As shown in Fig. 2, in our case, $F(s, r)$ was approximately estimated using the weighted average of $F_i(s)$ of all the grid cells within the circular ring of a certain width $r_c$ at distance $r$ from

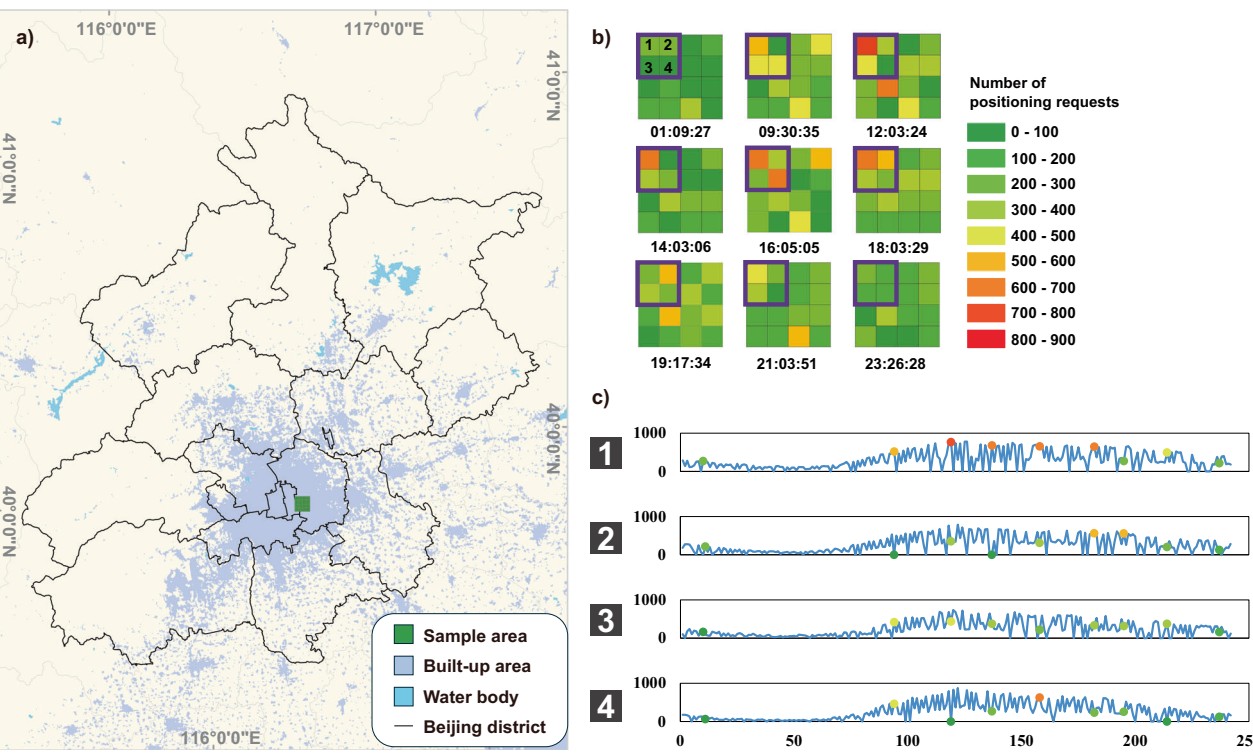

**Fig. 1 | Representation of population dynamics. a** An example map of a study area (the city of Beijing, China). The sample area in Green consists of 16 grid cells corresponding to (**b**). In this way, each grid cell in **b** has its spatial reference in (**a**). **b** Sequential and time-slicing mapping of mobile check-in data for the sample area in (**a**). The color in each grid cell represents the number of ambient population derived from mobile device datasets at a specific time spot (labeled below each image in **b**). **c** The time series recording the population fluctuations of the four grid

cells in the purple box in (**b**). The number labels on the left side of the series correspond to the number labels in the purple box of the first image in (**b**). The positions and values of the color points in the time series correspond to the spatial patterns of the four grid cells at the above nine time spots in (**b**). [Note: **a** is Powered by Esri. The built-up areas and water bodies on the base map of **a** were extracted from the European Space Agency's 2018 Land Cover–Climate Change Initiative data[61] (© 2017 ESA Climate Change Initiative – Land Cover, led by UCLouvain)].

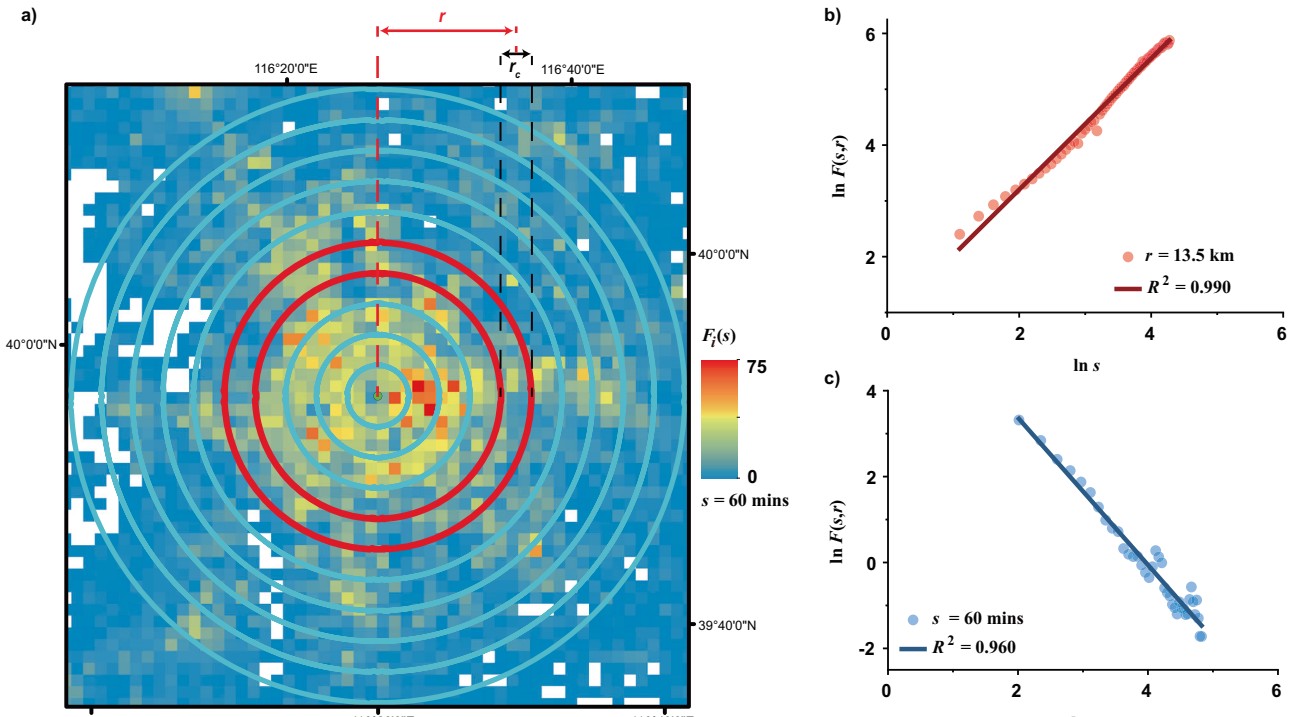

**Fig. 2 | Schematic diagram for measuring the mean population fluctuation. a** An example of spatially annular division of concentric zones in Beijing. For the six cases, the center of the concentric structure represents the main city center or metropolitan core determined by their urban or regional master plans (see Supplementary Note 2 and Figs. S1–6). Accordingly, Shenzhen has multiple main centers, while the other 5 cases have a single main center. The purpose of this division is to statistically calculate the mean population fluctuations, $F(s, r)$, at a given distance $r$ from the main urban center within a specific time interval $s$ (in this case, $s = 60$ min). For each circular ring with the width $r_c$, its distance from the center is regarded as the radial distance from the midpoint of its width to the center. The widths of circular rings for the six cases are different, which are determined considering the sizes of both the study areas and spatial grids. Specifically, the circular ring width of four Chinese cities and Greater Boston is 3 km (1.15 km × 1.15 km spatial grid for Chinese cities, nearly 1 km × 1 km spatial grid for Greater Boston, and over 50 km urban radius), and that of Milan is 0.5 km (235 m × 235 m spatial grid and about 10 km urban radius). **b** The temporal scaling of the mean population fluctuations $F(s, r)$ of the red circular region in (**a**). Each circular region in cities could possess such a temporal scaling, thus forming Fig. 3a. **c** The spatial scaling of the mean population fluctuations $F(s, r)$ of Beijing with the time interval $s = 60$ min. When changing the time interval $s$, spatial scaling of population fluctuations could accordingly change, thus forming Fig. 3b. [Note: **a** is Powered by Esri.].

the center (see "Methods" for details of calculation). Subsequently, we investigated the space-time scaling patterns of population fluctuations in four Chinese cities and conducted the same analysis in two other cities in Europe and North America, respectively, serving as validation (Fig. 3). The results show that population fluctuations in diverse urban systems across continents can be characterized by a set of scaling laws, which we define as:

$$F(s, r) \propto s^{\alpha(r)}, \alpha(r) > 0 \qquad (1)$$

$$F(s, r) \propto r^{d(s)}, d(s) < 0 \qquad (2)$$

where $\alpha(r)$ is the temporal scaling exponent of population fluctuations at a specific distance $r$ from the closest main urban center, and $d(s)$ refers to the spatial scaling exponent of population fluctuations within a given temporal scale $s$. These scaling laws maintain consistency despite alterations in the configuration of the annular structure and the corresponding distance sampling (Supplementary Note 3, Figs. S7 and S8, and Table S1).

Equation (1) defines the scaling law of population dynamics in the time domain, indicating that at a certain distance $r$ from the city center, the population fluctuations $F(s, r)$ increase in proportion to the increase of temporal scales $s$. Equation (2) characterizes the scaling law of population dynamics in the spatial domain, indicating that within a certain temporal interval $s$, the population fluctuations $F(s, r)$ decrease

in proportion to the increase of spatial distance $r$ from the closest city center. Equations (1) and (2) together reflect the spatiotemporal scale invariance of population dynamics, suggesting that we may not be able to find a specific spatial or temporal scale to characterize human motions.

It is noteworthy that only one obvious break in linearity among all the plots appears and this is in Shenzhen (see Fig. 3b, and similar situations also occur in our sensitivity analysis, as shown in Supplementary Fig. S8). A crossover point around $r = 25.5$ km (ln $r \approx 3.23$) indicates the spatial patterns of Shenzhen's population fluctuations exhibit two distinct scaling processes: one characterized by relative spatial homogeneity with a low slope close to the main urban area ($r \leq 25.5$ km) and the other by relatively high spatial heterogeneity with a high slope in the outer suburbs ($r > 25.5$ km). This is a common phenomenon that reflects different growth probabilities across locations or scales, leading to distinct scaling properties within multiscaling processes[40]. Each scaling process, when considered separately, conforms to the scaling law described in Eq. (2). Supplementary Tables S3 and S4 list the linear estimation results of all the scaling relations across the whole spatiotemporal scales for population daily fluctuations in the six focal cases (Fig. 3). The overall averages of the goodness of fit are around 0.994 for temporal scaling and 0.921 for spatial scaling (see Table 1). Such approximate linearity confirms the objective existence of the spatiotemporal scaling laws defined by Eqs. (1) and (2). Furthermore, the differences in these linear slopes indicate that the scaling exponents $\alpha(r)$ and $d(s)$ vary with spatial and

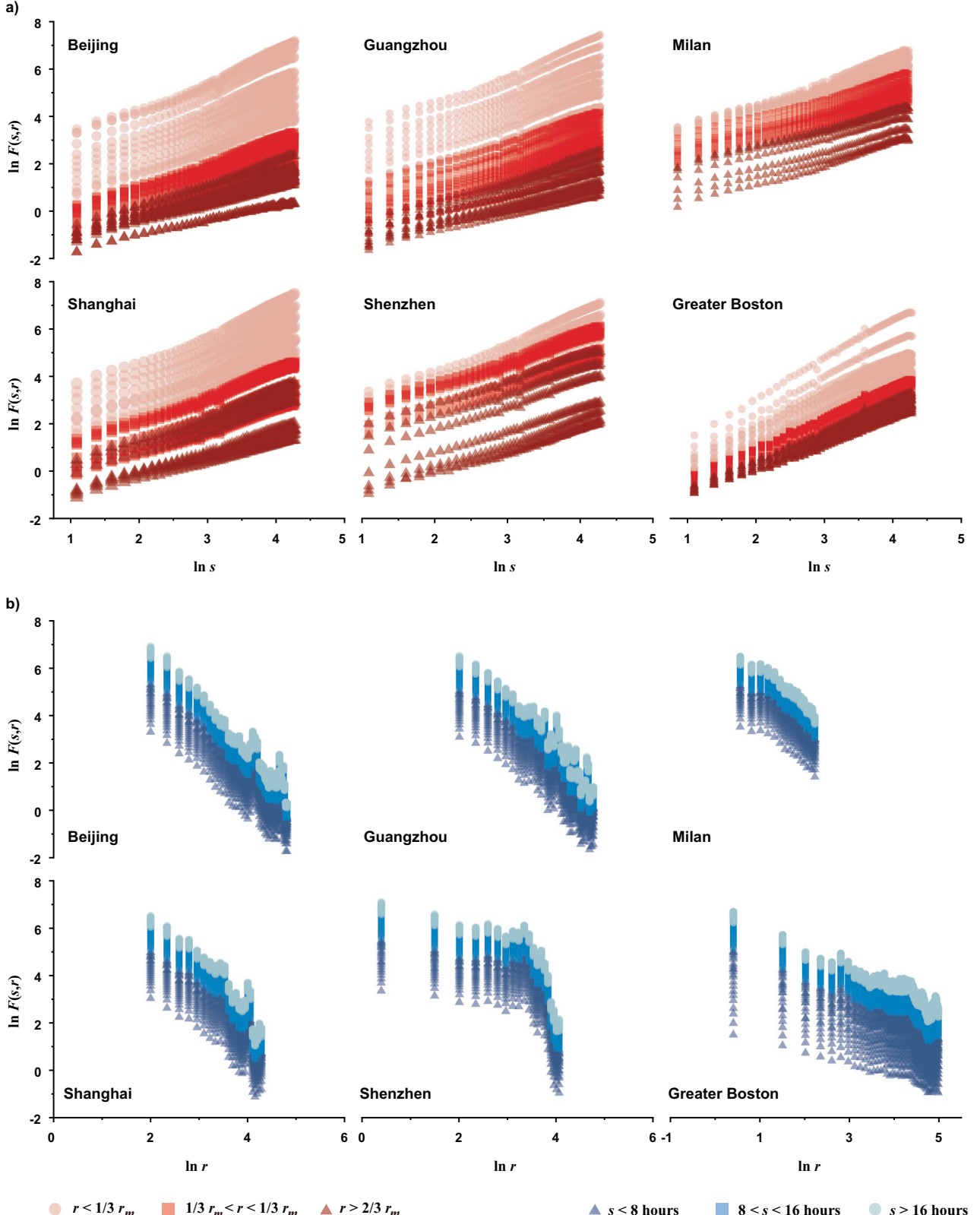

**Fig. 3 | Spatial and temporal scaling of urban population fluctuations. a** Double-logarithm plots of the mean population fluctuations $F(s,r)$ and temporal scales $s$ in terms of different spatial distance $r$ from the urban centers for the six focal cases. **b** Double-logarithm plots of the mean population fluctuations $F(s,r)$ and spatial distance $r$ from the urban centers in terms of different temporal scales $s$ for the six focal cases. $r_m$ refers to the maximum radius from the closest main urban center for each case.

**Table 1 | The ranges and averages of the goodness of fit, $R^2$, for the estimated temporal scaling exponent $\alpha(r)$ and spatial scaling exponent $d(s)$ of population fluctuations of the six focal cases**

| City name | $\alpha(r)$ | | $d(s)$ | |
|---|---|---|---|---|
| | $R^2$ | Avg. $R^2$ | $R^2$ | Avg. $R^2$ |
| Beijing | 0.973–0.999 | 0.994 | 0.943–0.963 | 0.947 |
| Shanghai | 0.985–0.998 | 0.992 | 0.888–0.923 | 0.900 |
| Guangzhou | 0.985–0.998 | 0.994 | 0.918–0.945 | 0.918 |
| Shenzhen | 0.988–0.997 | 0.992 | 0.834–0.922 | 0.893 |
| Milan | 0.988–0.996 | 0.992 | 0.943–0.961 | 0.956 |
| Greater Boston | 0.994–0.998 | 0.997 | 0.915–0.944 | 0.926 |

Avg.$R^2$ refers to the average of the goodness of fit, $R^2$. The data in this table is calculated based on Supplementary Tables S3 and S4.

temporal scales $r$ and $s$, respectively. This illustrates that the population fluctuations in cities or regions possess spatiotemporal heterogeneity of scaling, meaning that their scaling relationships could vary depending on different geographic locations or time scales.

## Heterogeneity of temporal scaling and regularity of spatial organization

While Eq. (1) reveals the existence and spatial heterogeneity of temporal scale invariance in population fluctuations at a relatively macro urban scale, it remains unclear whether this invariance universally exists at more micro scales and how this is expressed in terms of its spatial heterogeneity. Therefore, we investigated scaling behaviors of population fluctuations in the time domain at the grid scale. The double-logarithm plots of the mean fluctuation $F_i(s)$ and its counting time duration $s$ for any valid grid cells $i$ in the six focal cities are depicted in Fig. 4. These plots generally show strong linearity between the logarithmic functions $\ln F_i(s)$ and $\ln s$ of the mean fluctuations and

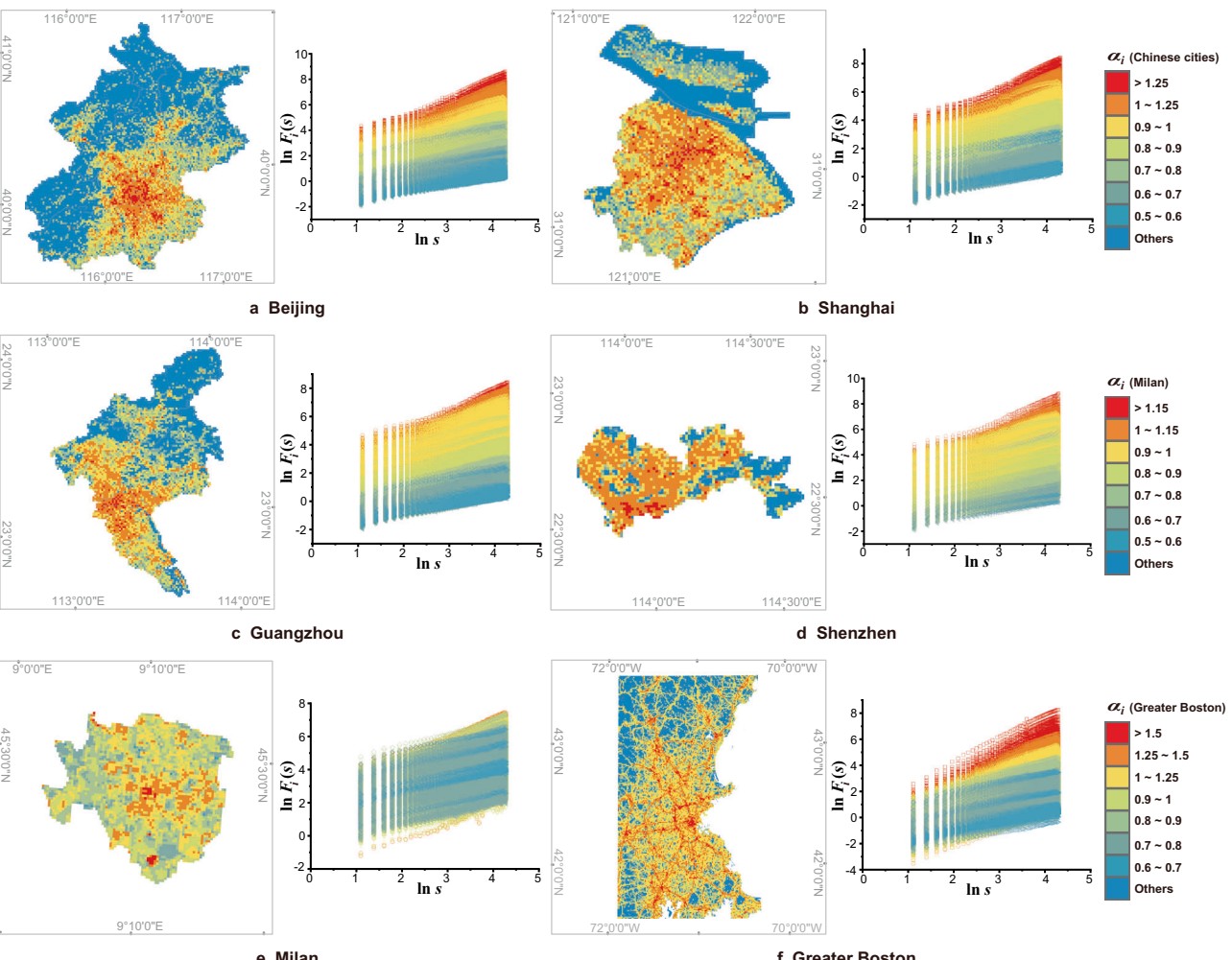

**Fig. 4 | Temporal scaling of the grid-scale population fluctuations $F_i(s)$ and spatial patterns of their scaling exponents $\alpha_i$ for the six focal cases. a–f** Double-logarithm plots of the mean population fluctuations $F_i(s)$ and temporal scales $s$ for each grid cell $i$ (right) and spatial distribution of temporal scaling exponents $\alpha_i$ at the grid scale (left) in the six focal areas are shown. Each curve represents a group of estimated mean fluctuations within the corresponding temporal scales for the time series of population dynamic fluctuation in a spatial grid. The exponents $\alpha_i$ can be estimated by calculating the slope coefficients of the curves. To clearly display the spatial structure of these urban systems, the areas of fBm with $\alpha_i > 1$ are divided into two levels (represented by red and orange colors) for those cities with $\alpha_i$ values generally less than 1.5 and three levels (represented by red, orange, and yellow

colors) for Greater Boston due to its $\alpha_i$ range exceeding 1.5. The levels of fBm are determined mainly based on the statistical properties of $\alpha_i$ and the $\alpha_i$ range of the city, such as the critical value $\alpha_i = 1.5$ (distinguishing the persistence and anti-persistence, see Supplementary Note 4) for Greater Boston, the critical value $\alpha_i = 1.25$ for Beijing, Shanghai, Guangzhou, and Shenzhen with their $\alpha_i$ values generally less than 1.5, and the critical value $\alpha_i = 1.15$ for Milan with its $\alpha_i$ values generally less than 1.3 (see Table 2). This figure generally shows spatial gradient patterns of $\alpha_i$ from the center to the periphery, indicating that the higher the value of $\alpha_i$ is, the closer a given area is potentially to the city centers. [Note: The left-side images of **a–f** are Powered by Esri.].

**Table 2 | The ranges of the estimated temporal scaling exponents, $\alpha_i$, and the ranges and averages of the goodness of fits, $R^2$, for the six focal cases**

| City name | $\alpha_i$ | $R^2$ | Avg. $R^2$ |
|---|---|---|---|
| Beijing | 0.530–1.519 | 0.913–0.998 | 0.991 |
| Shanghai | 0.523–1.491 | 0.935–0.999 | 0.990 |
| Guangzhou | 0.526–1.445 | 0.929–0.998 | 0.990 |
| Shenzhen | 0.592–1.463 | 0.969–0.996 | 0.990 |
| Milan | 0.549–1.293 | 0.948–0.997 | 0.989 |
| Greater Boston | 0.624–1.805 | 0.933–0.999 | 0.994 |

Avg.$R^2$ refers to the average of the goodness of fit, $R^2$.

time scales respectively. This indicates the geographically micro-scale universality of temporal scale invariance in population fluctuations, expressed as:

$$F_i(s) \propto s^{\alpha_i} \qquad (3)$$

where $\alpha_i$ denotes the scaling exponent of population fluctuation in the time domain for the grid cell $i$ and ranges from 0 to 2. Detailed information on the range of the estimated scaling exponents $\alpha_i$ and the goodness of fits, $R^2$, for the double-logarithm plots (Fig. 4) are provided in Table 2.

The value range of $\alpha_i$ ($0 < \alpha_i < 2$) suggests that the mean population variation could potentially increase with temporal scales in three different ways: a sub-linear increase with $0 < \alpha_i < 1$, a linear increase with $\alpha_i = 1$, and a super-linear increase with $1 < \alpha_i \leq 2$. Consequently, a higher $\alpha_i$ value implies more intensive long-term oscillations in population variation, and $\alpha_i = 1$ is a critical value used to distinguish between the dichotomous fractal time series models of fractional Gaussian noise (fGn) and fractional Brownian motion (fBm) (see Supplementary Note 4 and Fig. S12). Specifically, when $\alpha_i > 1$, the time series follow the non-stationary properties of fBm. When $\alpha_i < 1$, the series follow the stationary properties of fGn. The exponent $\alpha_i$ also has a mutually transformable relationship with the Hurst exponent and the spectral index[41]. Thus, it can characterize the long-range dependence of the time series[42] and indicate the phenomenon of "$1/f$ noise" (see Supplementary Note 4). If $\alpha_i > 0.5$, the series are long-term correlated. The higher the value of $\alpha_i$, the stronger the correlations in the series. In contrast, $\alpha_i < 0.5$ indicates the long-term anti-correlations of the series[43,44]. In our case, the estimated $\alpha_i$ values for the six focal cities generally range from 0.5 to 2.0. This indicates the long-term correlation (i.e., persistence) of dynamic human behavioral characteristics in urban systems.

Figure 4 and Table 2 suggest that the estimated scaling exponents $\alpha_i$ vary across locations within individual cities. This confirms the spatial heterogeneity of temporal scaling processes in population fluctuations, as similarly revealed by Eq. (1). After classifying the grid cells within these cities based on the $\alpha_i$ values, the spatial arrangement of $\alpha_i$ (Fig. 4) shows patterns similar to urban structures as determined by their urban master plans (Supplementary Note 2 and Figs. S1–6) and the spatial regularity of a general distance-decay organization from the center to the periphery. Specifically, the areas with high $\alpha_i$ values are primarily located in planned main urban areas, the core areas of new towns and counties, or important transportation corridors, while the areas with low $\alpha_i$ values are predominantly distributed on the periphery of these cities and regions. Typically, the higher values of $\alpha_i$ (i.e., $\alpha_i > 1.25$ for Beijing, Shanghai, Guangzhou, and Shenzhen; $\alpha_i > 1.15$ for Milan; $\alpha_i > 1.5$ for Greater Boston) essentially correspond to key functional areas or development hot spots, such as the central business districts, the cities' historic or development cores, as well as important

economic, technological, and industrial functional areas and nodes in the urban network (Supplementary Note 5 and Figs. S13–18).

The concentration of higher $\alpha_i$ values in main urban areas is potentially formed by a high level of population mobility in or around urban centers, key functional areas, or development hotspots. This is especially due to higher population density (which potentially creates more population movements in the unit area) and highly frequent visitations caused by regular commuting and relatively mixed land uses but usually with the leading functions of manufacturing, commerce, or public services (which may create directional population flows). This potentially generates more significant global variation in population dynamics but with smoother fluctuations as the short-term population changes are slight compared with the large base of population movements. In contrast, a low level of population mobility, potentially due to low population density, limited public transport coverage, singular economic structures focused on primary industries, or restricted and limited urban functions, makes the population relatively stable, without the obvious rise–fall in oscillations, thereby resulting in lower $\alpha_i$ values scattered around the outskirts of the city.

## Spatiotemporal gradients of scaling exponents

The spatial regularity of higher $\alpha_i$ values concentrating in main urban centers and core functional areas and gradually decreasing outwards (Fig. 4) suggests that temporal scaling exponents $\alpha_i$ could statistically possess a spatial gradient from the center to the periphery. Figure 5 exhibits the spatial variation of the mean $\alpha_i$ values with regard to the distance $r$ from main urban centers, based on estimating the logarithmic spatial gradient of temporal scaling exponents, that is:

$$\alpha(r) = -b \ln r + \alpha_1 \qquad (4)$$

where $\alpha(r)$ has the same meaning as defined in Eq. (1), $b$ reflects the gradient decreasing rate, and $\alpha_1$ is a constant coefficient that can be interpreted as the theoretical scaling exponent near the city center ($r = 1$).

A simple integral after combining the derivation of Eq. (1) with Eq. (4) yields not only the spatiotemporal scaling laws described as Eqs. (1) and (2), but also a logarithmic temporal gradient of spatial scaling exponents ("Methods"), expressed as:

$$d(s) = -b \ln s + d_1 \qquad (5)$$

where $d(s)$ and $b$ have the same meanings as defined in Eqs. (2) and (4) respectively, and the constant coefficient $d_1$ can be regarded as the theoretical scaling exponent of population fluctuations in one temporal unit, that is, $F(r) \propto r^{d_1}$ when $s = 1$. The temporal gradient of $d(s)$ can be empirically verified through the estimation of Eq. (2), as shown in Supplementary Fig. S19.

Equations (4) and (5) complete the expression of scaling exponents defined in Eqs. (1) and (2), thereby revealing the regularity of spatiotemporal complexity in population dynamics. The scaling relationship of population fluctuations is not governed by a single scaling exponent but varies according to temporal scales and geographical locations. However, this variation in scaling exponents follows a logarithmic spatiotemporal gradient with a consistent gradient rate, $b$. Only when the temporal measurement scale or geographical location is specified can a specific spatial or temporal scaling relation of population fluctuations be determined. Above all, the spatiotemporal scaling model composed of Eqs. (1) and (2) when integrated with Eqs. (4) and (5) quantifies the scaling processes and spatiotemporal sensitivity of population dynamics. Consequently, this model is capable of reproducing and predicting the spatiotemporal patterns of urban population fluctuations in cities and regions (see Fig. 6 for reproduction results and Supplementary Note 3 and Fig. S9 for prediction

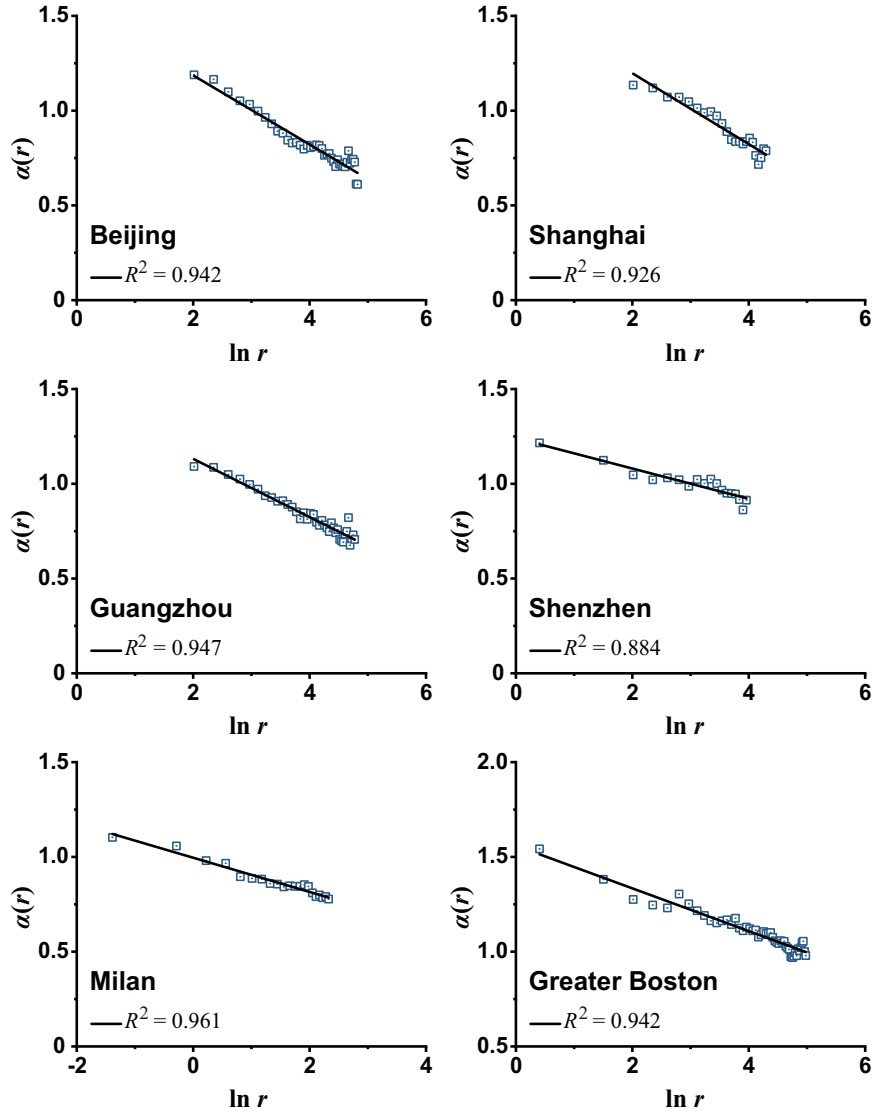

**Fig. 5 | Spatial gradient of scaling exponent $\alpha(r)$ values from main urban centers to the periphery for the six focal cases.** The spatial circular division of the six areas, based on which temporal scaling exponents $\alpha(r)$ are estimated, is consistent with that described in Fig. 2.

results of the four cases not used for parameter estimation) by using the estimated parameters (Table 3).

### An urban allometry of population dynamics

As we previously speculated, the spatial patterns of population dynamics may be related to population density and functional attractiveness, which are the main factors affecting mobility[5,17], while also showing certain spatial gradients[17,32]. To investigate these relationships, we characterized both population density and functional attractiveness in terms of urban density in our case. The density of points of interest (POI) reflects the degree of aggregation of urban functions in a region, while higher POI density represents a higher attractiveness of urban functions to some extent. Typically, the spatial distribution of urban density is simulated using either the negative exponential model or the inverse power model[45], which represent the urban density $\rho(r)$ of a location at distance $r$ from the city center (see "Methods" for details). Although there is controversy surrounding the choice and applicability of these two models, Batty and Kim[38] argued that the most rational notion of urban density is to focus on the availability of urban space for development, suggesting that the inverse power model is more appropriate. In line with this notion, the inverse power model of urban density and the spatiotemporal

scaling law of population fluctuations yield a spatiotemporal allometric relationship through a simple derivation (Eq. (16)):

$$F(s,r) \propto \rho(r)^{A(s)} \tag{6}$$

where $A(s) = \frac{d(s)}{\beta} = \frac{-b\ln s + d_1}{\beta}$ is the allometric exponent characterizing how the degree of population mobility scales with the base population or urban functions.

We validated the derived allometry in Eq. (6) using empirical data on population mapping and points of interest (POI) (see "Data description" in "Methods"). Here we present the double-logarithm relationships of mean population fluctuations and two different urban densities across various temporal scales for six focal cases (Fig. 7). The sensitivity of these relationships to measurement schemes was examined through four sets of new observations (Supplementary Note 3 and Figs. S10 and S11). The approximate linearity suggests a degree of urban allometry in population dynamics with respect to urban density. Supplementary Tables S5–7 provide a detailed summary of the allometric exponents, goodness of fit, and a comparison of the two different urban densities. For further details, refer to Supplementary Note 3 and Table S2, which summarize the four groups used for sensitivity analysis. These results indicate that population

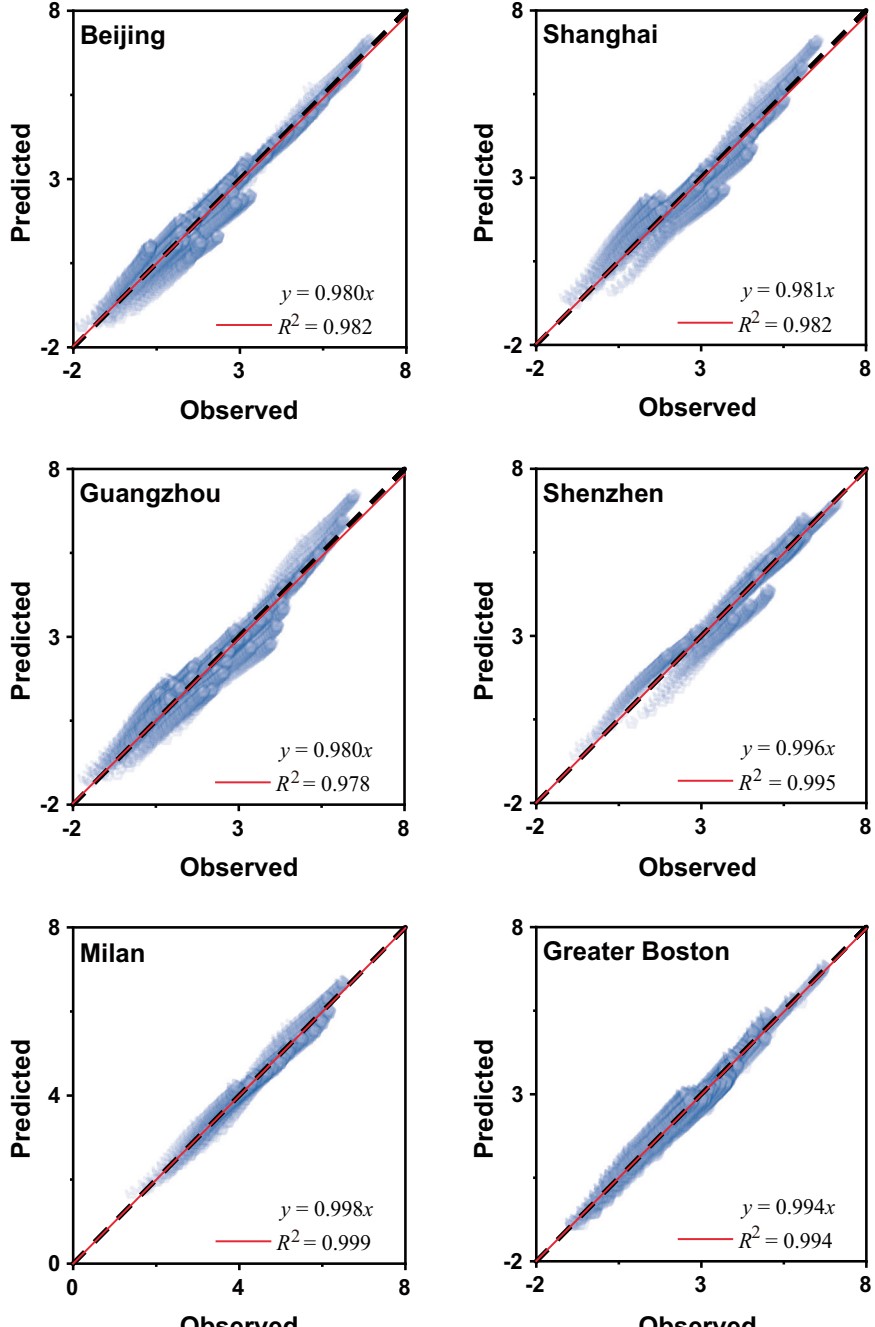

**Fig. 6 | Plots of the observed versus predicted population fluctuation *F(s,r)* expressed in a logarithmic manner for the six focal cases.** The dashed line represents a perfect prediction for the observation (i.e., *y = x*). The closer the slope coefficient of plots (fitted by the red lines) is to 1, the better the predicted results are to the observed. The logarithmic expression ensures the comparability across time and space scales in a plot by eliminating dimensional inconsistencies.

fluctuations generally exhibit better urban allometry with the density of POI (with average goodness of fit above 0.87 for the six cases) compared to population density in these cases. This difference can be attributed to the relatively low spatial variations in population density near city centers in cities like Milan and Boston, causing deviations from the inverse power distribution around city centers (see Supplementary Fig. S20). Consequently, the stronger urban allometry between population fluctuations and POI density underscores the influence of urban functions on population dynamics, assuming a similar population base for these cities. In contrast, the allometries of population fluctuations with both urban densities for all four Chinese cities are more consistent. It is also worth noting that the distributions

of both types of urban densities (Supplementary Fig. S20) in Shenzhen similarly exhibit two scaling processes relating to its population fluctuations (Fig. 3b). This further demonstrates their intimate correlations.

The allometric relationship expressed by Eq. (6) reveals that more densely populated areas or areas with more urban functions (as indicated by urban POI density in our case) within a city can contribute to greater population mobility in a proportional manner. This is because a larger population base has the potential to generate more population movements, and more urban functions can result in increased visitations. Furthermore, Eq. (6) suggests that the urban allometry of population fluctuations with urban density is also influenced by the

**Table 3 | The estimation of the parameters, $b$, $\alpha_1$, $d_1$, and the constant, for the spatiotemporal scaling models of the six focal cases**

| City name | $b$ | $\alpha_1$ | $d_1$ | Const |
|---|---|---|---|---|
| Beijing | 0.203 | 1.717 | −1.317 | 4.180 |
| Shanghai | 0.156 | 1.559 | −1.532 | 4.843 |
| Guangzhou | 0.165 | 1.566 | −1.486 | 4.943 |
| Shenzhen | (0.077,0.203) | (1.277,1.764) | (−0.099, −5.190) | (1.692,19.264) |
| Milan | 0.091 | 1.005 | −1.095 | 3.260 |
| Greater Boston | 0.085 | 1.605 | −0.508 | 0.342 |

The parameters are defined in Eq. (12), and the values in this table are estimated based on the linear relationships plotted in Supplementary Fig. S19. The first value in each parenthesis represents the estimated result corresponding to $r \leq 25.5$km for Shenzhen, and the second value represents the estimated result corresponding to $r > 25.5$km for Shenzhen.

temporal scale, with the allometric exponents exhibiting a positive logarithmic temporal gradient (Supplementary Fig. S21). This finding provides a potential means to predict population dynamics using urban indicators. As demonstrated in a derivation (see "Methods"), urban density and temporal scaling exponents follow a logarithmic relationship (Supplementary Fig. S22). This indicates that when the population size or numbers of urban functions reach a certain threshold (e.g., for the four Chinese cities, the estimated population density is around 5000 people/km²), adequate population and accessible urban functions give rise to non-stationary population dynamics, resulting in a super-linear trend [i.e., fBm with $\alpha(r) > 1$]. In contrast, a shortage of population or limited access to urban functions can dampen population mobility, leading to a sub-linear trend [i.e., fGn with $\alpha(r) < 1$] due to reduced demand for travel or visitations.

## Discussion

Human mobility is a critical component of urban systems, characterized by a complex array of trajectories, flows, and population fluctuations across various locations. These dynamics, which follow the innate periodicities of social activities and human life on a daily basis, faithfully embody their regularities[1] and reflect the mechanisms driving socio-economic operations within cities, pinpointing critical aspects of urban development[3,46]. However, despite their significance, these dynamics, especially population fluctuations that represent the urban pulse, have not been investigated in depth, perhaps due to the complexities involved in extracting and succinctly characterizing such time-dependent phenomena. In our study, we analyzed large-scale mobile device data containing over 800 million records from multiple cities and regions around the world to investigate population fluctuations across space and time. We found that universal scaling laws govern the spatiotemporal dynamics of population fluctuations in urban systems regardless of their sizes, densities, or geographic locations. Notably, our findings reveal that urban phenomena also exhibit temporal scale invariance (Figs. 3a and 4) and underscore the relevance of scaling in both space and time (Fig. 5 and Supplementary Fig. S19). We also discovered an allometric scaling relation that links these dynamic fluctuations with static urban density (Fig. 7). Our findings, robustly validated across diverse datasets and using varied measurement approaches (Supplementary Note 3), not only provide a distinctive framework capable of characterizing, interpreting, and predicting population dynamics, but also advance the fractal analysis of urban systems by integrating space and time. This study is among the first to explore these combined aspects.

When using mobile device data to represent human dynamics, a common issue of data biases arises. Currently, there is no simple or standardized solution to address this problem, especially due to the lack of ground-truth estimates and the limited availability of similar dynamic data with comparable spatiotemporal information for cross-validation[47–49]. To maximize the authenticity of our findings, we utilized multi-source data that differs in user coverage, collection method (i.e., through cellular towers or global positioning system), time and location, spatiotemporal resolution, and usage patterns (see "Data description" in "Methods"). The consistency of the resultant regularities indicates the generality and reliability of our discovered laws on urban population fluctuation, which were minimally affected by different sampling methods or biases.

The phenomenon of "$1/f$ noise," identified in diverse natural systems, implies self-organized criticality (SOC)[50,51]. While it has been proposed to characterize evolutionary urban processes in this way[52], its temporal manifestation and the interpretation of urban SOC remain ambiguous[53]. However, our discovery of the spatiotemporal scaling law for urban population dynamics implies such criticality. A consistent gradient relating temporal scaling exponents to structure, density, and function (Figs. 4 and 5, Supplementary Figs. S1–6, Figs. S13–18 and S22) across six study areas reveals an urban SOC process spatially associated with "$1/f$ noise." This not only addresses the long-standing speculation that SOC with a signature of "$1/f$ noise" could relate to the spatial structure of matter[50], but also quantifies these spatial relationships for urban systems.

Our findings demonstrate that the spatial distributions of dynamic human activities in cities exhibit a mixed organizational structure. At the microscale, the dynamics show a patchy cluster pattern radiating unevenly outward from city centers (Fig. 4). This reflects fundamental growth frameworks shaped by transportation corridors that drive sprawl beyond cores. It also aligns with the sector and multiple nuclei models by revealing functional clusters along such routes outside city centers[54,55] (Supplementary Note 5 and Figs. S13–18). Despite exhibiting irregularly polycentric structures (Supplementary Figs. S1–6), at the macroscale (municipal/metropolitan levels), population patterns, whether centered on single or multiple cores, statistically conform to Burgess's concentric zone model[56], which is widely applicable to large metropolitan areas, particularly those that have evolved during the industrial era. We thus quantitatively characterize the temporal variations corresponding to urban structures and locations, resulting in intuitive space-time spectra that unravel the underlying spatiotemporal regularities of human activity patterns and their relation to *de facto* urban spatial organization.

Our proposed models hold pragmatic significance. The spatiotemporal scaling analysis of big human mobility data effectively identifies the actual spatial organization of cities, fundamentally reflecting underlying activity patterns. This offers guidance for optimizing the placement of urban POI, such as public facilities, retail locations, and advertising spaces like posters and billboards, to enhance livability and accessibility, as well as to boost tourism development. Space-time spectra also enable the detection, measurement, and assessment of sprawl through a human mobility lens. Comparing these spectra with current urban development can intuitively and objectively evaluate the rationality of the structural-functional layout, addressing issues such as imbalanced growth. Additionally, characterizing the duality of fBm and fGn in cities provides new perspectives on links between population and urban functions, which facilitates objective urban-rural analysis and informs the assessment of urbanization. The identification of activity hotspots further aids in pandemic response by enabling targeted prevention in high-risk areas. Our findings also inform nonlinear transportation planning by considering infrastructure needs relative to fluctuating population distributions.

While our findings shed light on urban dynamics, they also highlight the need for further research. For example, a deeper investigation into city polycentricity and daily human activity rhythms may be needed to improve predictions of population fluctuations. However,

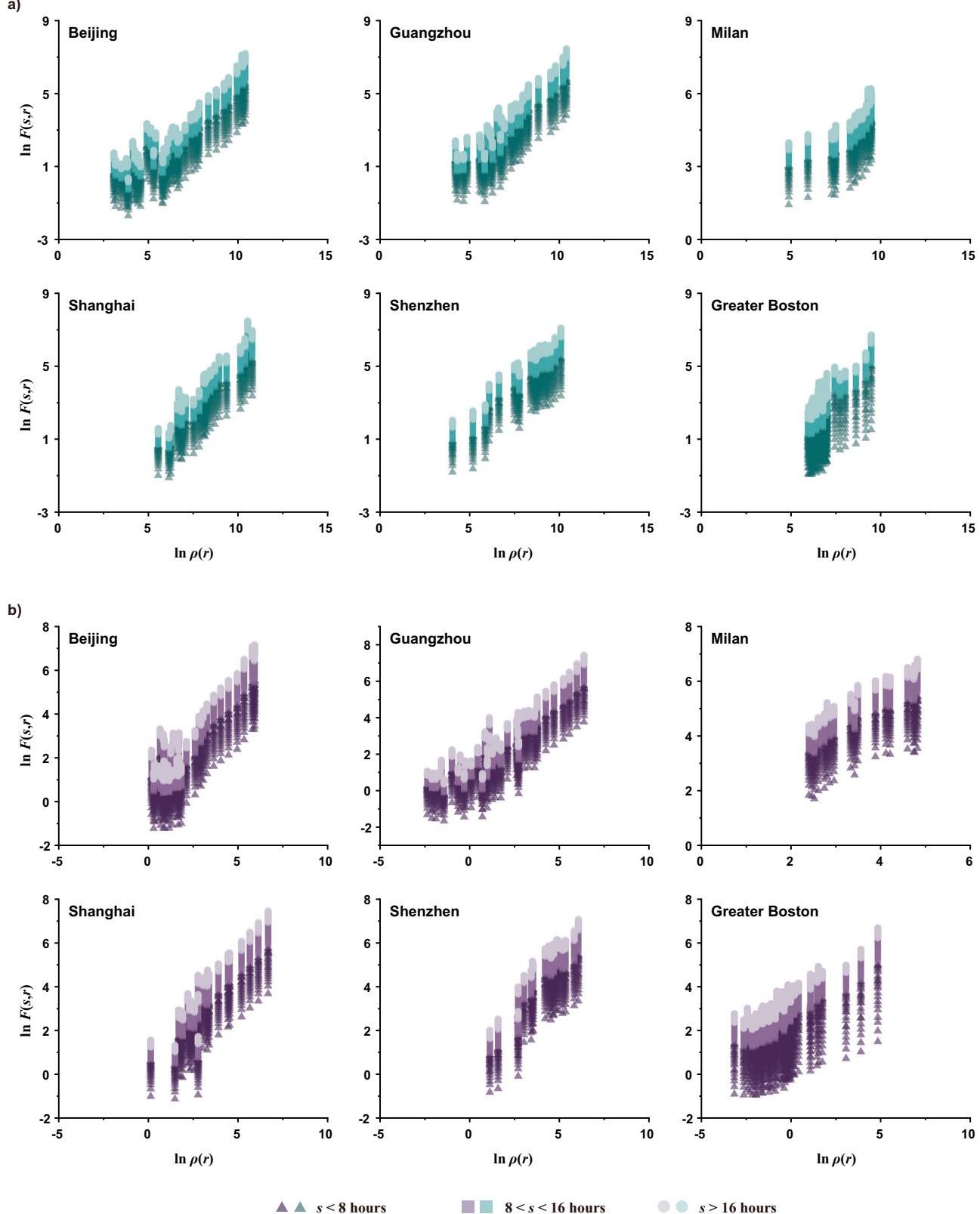

**Fig. 7 | Urban allometric relations between the mean population fluctuations $F(s,r)$ and urban densities $\rho(r)$. a** Double-logarithm plots of the mean population fluctuations $F(s, r)$ and population density $\rho(r)$ in terms of different temporal scales $s$ for the six focal cases. The population densities of all six areas are estimated based on the data from WorldPop (see "Data description" in "Methods" for more details). The estimation of the slope coefficients in **a** yields the allometric exponents $A(s)$ of population density. **b** Double-logarithm plots of the mean population fluctuations

$F(s, r)$ and the density $\rho(r)$ of points of interest (POI) in terms of different temporal scales $s$ for the six focal cities. The densities of POI of the four Chinese cities are estimated by using the data crawled from Amap, while the estimation of POI densities for Milan and Greater Boston is based on the data from OpenStreetMap (see "Data description" in "Methods" for more details). The estimation of the slope coefficients in **b** yields the allometric exponents $A(s)$ of POI density.

our research still faces data-related and methodological challenges and uncertainties within the current paradigm of complexity. These include insufficient data resolution for a thorough examination of local scaling, as well as issues of statistical significance, computational complexity, comparability, and practical interpretation when examining temporally multi-scaling processes. Although our focus has been on daily population dynamics, studying intercity population movements over longer timescales (e.g., monthly or yearly) remains essential for understanding broader migratory patterns. Bridging the macroscopic and microscopic scales of urban complexity is a significant endeavor. Future work should aim to consolidate these dimensions to advance transportation, urban planning, and policymaking.

## Methods

### Data description

To increase the generalizability of our findings in this study, we considered the diversity in the various cases (defined in Supplementary Notes 1 and 2), the variety of data sources and types, and their time taken to acquire these. Our grid-based population time series are generated from three different mobile device datasets, including check-in data for four Chinese cities, records of mobile phone network connections for the city of Milan, and GPS data for Greater Boston.

The check-in data for four Chinese cities are derived from Tencent's *Xingyun* map (https://xingyun.map.qq.com/), which anonymously collects user locations through mobile apps using Tencent location services, thus covering a huge Chinese user group. We scraped the check-in records over 10 consecutive weekdays in 2018 (April 16–27, 2018) when there were no public holidays. Our crawled data (a total size of 33.6 GB) contains numerous positioning requests in a global geographical coordinate system (with a spatial resolution of 0.01 degrees) at discrete time spots. The crawling frequency is about 6 min on average.

Data for Milan were obtained from an open dataset created by Barlacchi et al.[49] based on "Telecom Italia Big Data Challenge", a program spearheaded by Telecom Italia, providing geo-referenced and anonymized datasets under the Open Database license. The data contains about 57.4 million records of mobile phone users within 10 consecutive weekdays in 2013 (November 04–15, 2013). These records quantify the activities proportional to the true amount of Internet connections in given grid cells (about 235 meters by 235 meters) of Milan within a given time interval (10 min).

Data for the Greater Boston area was sourced from a large dataset of GPS pings from mobile devices provided by SafeGraph, a company who collects and markets location-based data associated with human mobility. The original dataset anonymously records key features relating to individual movement behaviors during two consecutive weeks of working days in 2020 (February 03–14, 2020) at the scale of census block group. These records were finally processed to generate grid-based (about 1 km by 1 km) population time series data.

As the minimum temporal scales of the three data sets are unequal, to ensure the consistency of temporal intervals and the equivalence of data records within each temporal interval, we have taken the interval of 20 min as the minimum temporal unit of the time series. Within this interval, the records were counted and accumulated as the nominal value of population. As a result, the data on any one day contain 72 time spots and the total time series of 10 weekdays comprises 720 time spots.

In addition, data relating to population density and POI are also used in our study. Population density data (from WorldPop) estimates the number of people in a grid-cell at a resolution of 3 arc seconds (approximately 100 meters at the equator) for China in 2018, Italy in 2013 and the United States in 2020, which is temporally consistent with the years when the three mobile device datasets were acquired. To maximize the accuracy of POI acquisition while ensuring data availability, the POI data for the city of Milan and the region of Greater Boston were directly downloaded from OpenStreetMap (https://www.openstreetmap.org/), while those for Chinese cities were collected from Amap (https://www.amap.com/) due to the significant lack of Chinese POI records in OpenStreetMap. In a similar vein, the years of POI data align with that of mobile device data for the six cities.

### Grid-based time series of population fluctuations

We study how the number of people in a place changes over time at a fine spatial scale. For spatially fine-grained population variations, although their spatial patterns are dynamic and ever-changing, they are conventionally represented in cross-section at a certain time point using a two-dimensional display (e.g., Fig. 1a, b). However, such a manifestation of time slicing (e.g., Fig. 1b) inevitably loses the expression of their dynamics. Here we develop grid-based time series that can capture all of the related spatiotemporal patterns of population fluctuations (e.g., Fig. 1b, c). Specifically, for each grid with human activities (time-dependent population records) in a city (e.g., the grid labeled as "1" inside the purple box in Fig. 1b), we associated its geographic location (i.e., the spatial coordinates corresponding to the position of the "Grid 1" in Fig. 1a) with its time series (i.e., the first time series marked as "1" in Fig. 1c) thus recording the population fluctuations that occurred within this grid cell during a given time period.

### Temporal scaling analysis

In this study, we used the method of DFA, initially introduced by Peng et al.[37], to define the concepts relating to population fluctuations and conduct the scaling analysis in the time domain. We chose this method because of its broad applications, its alignment with our research objective of analyzing the scaling effects of temporal dynamics, and its ability to succinctly characterize the dynamics and long-term correlations of time-varying processes in a straightforward and easily interpretable manner, without requiring complex transformations. In addition, due to its detrending process, DFA can eliminate a false statistical estimation of long-range correlation potentially caused by strong trends in non-stationary time series, and thus it can be applied to both stationary and non-stationary time series with unknown trends and noise[44,57].

Given a time series, $\{x_i(k)\}_{k=1}^{N}$, of length $N$ recorded in the $i$-th grid cell, the DFA procedure mainly consists of the following five steps[37,44,57]:

First, the global profile is determined by calculating the cumulative sum of the records and subtracting the mean using Eq. (7); an example of the creation of a profile is shown in Supplementary Fig. S23a:

$$Y_i(m) = \sum_{k=1}^{m} \left[ x_i(k) - \langle x \rangle \right], m = 1, \ldots, N \tag{7}$$

where $Y_i(m)$ and $\langle x \rangle$ denote the profile and the average of the time series $\{x_i(k)\}$ of the $i$-th grid, respectively. The subtraction is unnecessary as the later detrending step can produce the same effect.

Second, given a temporal scale $s$, the global profile $Y_i(m)$ is divided into $N_s = \text{int}(N/s)$ non-overlapping intervals with an equal length. By changing the length of the temporal scale $s$, different segmentation results can be achieved (Supplementary Figs. S23b and S23c). As the total length $N$ may not be exactly divided by the temporal scale $s$, a short part at the end of the profile will remain uncovered; therefore, the same procedure is repeated starting from the opposite side of the record. Consequently, $2N_s$ intervals are obtained.

Third, for each of the total $2N_s$ intervals under a temporal scale $s$, the local trend is estimated using a linear least-square fit of the data. Subsequently, the detrended profile $Y_s(m)$ is considered as the difference between the original profile $Y_i(m)$ and its local fits of each

interval $p_v(q)$ as Eq. (8):

$$Y_s(m) = Y_i(m) - p_v(q), \text{for } v = 1, \dots, 2N_s \text{ and } q = 1, \dots, s \qquad (8)$$

where $p_v(q)$ represents the fitting polynomial for the $v$-th interval, and $q$ denotes the order of time points within each interval. For simplicity, a linear regression is normally used to estimate $p_v(q)$. However, this may cause an inaccurate local trend when the temporal scale $s$ is large, leading to serious fluctuation in the subsequent linearisation of power law relations. In this study, we adopted an improved approach called temporally weighted DFA[57] to increase the local fitting accuracy. Supplementary Fig. S23d shows the process of local detrending for $s = 50$.

Fourth, for the $v$-th interval of the detrended profile $Y_s(m)$, the variance $F_s^2(v)$ is calculated using the two following equations, corresponding to the two directions in which the profile is divided in the second step:

$$F_s^2(v) = \frac{1}{s}\sum_{q=1}^{s} Y_s^2[(v-1)s + q], \text{for } v = 1, \dots, N_s \qquad (9)$$

and

$$F_s^2(v) = \frac{1}{s}\sum_{q=1}^{s} Y_s^2[N - (v - N_s)s + q], \text{for } v = 1, \dots, N_s \qquad (10)$$

Fifth, by averaging the variances $F_s^2(v)$ of all $2N_s$ intervals and taking the square root, the mean fluctuation (also referred to as the DFA fluctuation function) $F_i(s)$ is obtained according to Eq. (11). For a time series with long-range correlations, its mean fluctuation $F_i(s)$ and the corresponding temporal scale $s$ follow the power law relation as Eq. (3):

$$F_i(s) = \left[\frac{1}{2N_s}\sum_{v=1}^{2N_s} F_s^2(v)\right]^{1/2} \qquad (11)$$

In addition, based on the estimation of the mean fluctuation in each grid cell, we can approximately estimate the mean population fluctuation $F(s,r)$ at the distance $r$ from the main urban center during a given time scale $s$, by calculating the area-weighted average, $\sum_{i=j\sim n} A_i F_i(s)/\sum_{i=j\sim n} A_i$, of population fluctuations, $F_j(s) \sim F_n(s)$, for the grid cells $j \sim n$ intersecting the annular region (Fig. 2a) considered at the distance $r$ from the main urban center by the area of $A_i(i \in [j, n])$. This scheme accounts for the potential segmentation of grid cells by the ring structure by calculating the proportion of each grid cell's area that overlaps with each annular region as a weighting factor. To assess the influence of such segmentation, we conducted a sensitivity analysis by altering the configuration of the annular structure and examining its effect on the results (Supplementary Note 3).

**Derivation of spatiotemporal scaling laws**

Based on the combination of the derivation of Eq. (1) with Eq. (4), we can simply derive the relationship among the mean fluctuation $F(s,r)$, temporal interval $s$, and the distance $r$ from the main urban center, as expressed in Eq. (12):

$$\frac{d \ln F(s,r)}{d \ln s} = \alpha(r) = -b \ln r + \alpha_1$$

$$d \ln F(s,r) = -b \ln r\, d \ln s + \alpha_1 d \ln s$$

$$\int d \ln F(s,r) = \int -b \ln r\, d \ln s + \int \alpha_1 d \ln s$$

$$\int d \ln F(s,r) = -b \ln r \int d \ln s + \alpha_1 \int d \ln s$$

$$\ln F(s,r) + C_1 = -b \ln r (\ln s + C_2) + \alpha_1(\ln s + C_3)$$

$$\ln F(s,r) = -b \ln s \ln r + d_1 \ln r + \alpha_1 \ln s + \text{Const} \qquad (12)$$

where $C_1$, $C_2$, and $C_3$ are constants, $d_1 = -bC_2$, and $\text{Const} = \alpha_1 C_3 - C_1$. Equation (12) can be reorganized into Eq. (13) in terms of the dominant independent variable $r$.

$$\ln F(s,r) = (-b \ln s + d_1) \ln r + C \qquad (13)$$

where $C = \alpha_1 \ln s + \text{Const}$. After the inverse logarithmic transformation of Eq. (13), we can yield Eq. (2) with the logarithmic gradient expression of the scaling exponent $d(s)$ expressed as Eq. (5).

**Urban density models**

Two basic mathematical functions are commonly used to simulate the distance decay of urban density from the city center to the outer edge[32,38,45]. One was initially introduced by Clark[58] and is expressed by the negative exponential function:

$$\rho(r) = \rho_0 e^{-\mu r} = \rho_0 e^{-r/r_0} \qquad (14)$$

where $\rho(r)$ represents the urban density at distance $r$ from the city center ($r = 0$), $\rho_0$ is a constant coefficient indicating the central population density $\rho(r=0)$, $\mu = 1/r_0$ reflects the gradient decreasing rate, and $r_0$ quantifies the characteristic radius of the population distribution, which explicitly denotes the mean travel distance to the city center[59].

The other was proposed by Smeed[60] and is expressed by the inverse power function:

$$\rho(r) = \rho_1 r^\beta, \beta < 0 \qquad (15)$$

where $\rho(r)$ and $r$ have the same physical meaning as in Clark's model, $\rho_1$ is the constant of proportionality but without representing the central density ($r = 0$), and $\beta$ denotes the scaling exponent of the density distribution with the role of elasticity between the percentage change in density and the percentage change in distance. According to the mathematical definition of density and the theory of allometric growth, $\beta$ is also equal to $d - D_f$, where $d$ refers to the Euclidean dimension of the embedding space and $D_f$ represents the radial dimension of urban form. Thus, Smeed's model is strongly related to urban fractal theory, and its exponent $\beta$ measures the extent to which space is filled[38].

**Derivation of urban allometry**

When the spatial pattern of urban density follows an inverse power function, Eq. (15) can simply be transformed into the expression: $r^\beta = \rho(r)/\rho_1$. By associating this transformation with Eq. (2), we derive the allometry described by Eq. (6), which connects population fluctuation with urban density through the following steps:

$$F(s,r) \propto r^{d(s)} \propto (r^\beta)^{\frac{d(s)}{\beta}} \propto \left(\frac{\rho(r)}{\rho_1}\right)^{\frac{d(s)}{\beta}} \propto \rho(r)^{\frac{d(s)}{\beta}} \qquad (16)$$

**Derivation for the logarithmic density gradient of $\alpha(r)$**

Based on Eqs. (4) and (15), we can simply deduce the logarithmic relationship between the temporal scaling exponent $\alpha(r)$ and the urban density $\rho(r)$ through their common variable $r$, as expressed in

Eqs. (17) and (18):

$$\ln r = \frac{\alpha(r) - \alpha_1}{-b} = \frac{\ln \rho(r) - \ln \rho_1}{\beta} \tag{17}$$

$$\alpha(r) = -\frac{b}{\beta}\ln \rho(r) + \frac{b}{\beta}\ln \rho_1 + \alpha_1 \tag{18}$$

where the parameters, $\alpha_1$, $b$, $\rho_1$, and $\beta$, are estimated coefficients and constants for a city.

## Reporting summary

Further information on research design is available in the Nature Portfolio Reporting Summary linked to this article.

## Data availability

The original data from Tencent and SafeGraph used in this study are not publicly available due to licensing and privacy concerns. The pre-processed georeferenced time series supporting the findings of this study, which constitute a reorganization of the raw data, still require permission from the respective data providers for limited sharing. Data for Milan were obtained from an open dataset created by Barlacchi et al.[49] [https://doi.org/10.1038/sdata.2015.55]. Population data were derived from WorldPop [https://www.worldpop.org/datacatalog/], which provides open spatial demographic data. Points of interest (POI) data for Chinese cities were collected from Amap [https://www.amap.com/], while POI data for Milan and Greater Boston were downloaded from OpenStreetMap [https://www.openstreetmap.org/]. Due to licensing concerns, the crawled data are not publicly available. Other data supporting the findings of this study are available in the Source data. Source data are provided with this paper.

## Code availability

The code for data pre-processing (developed in Python and MATLAB), the temporally weighted detrended fluctuation analysis (developed in MATLAB), and fluctuation prediction (developed in MATLAB) are available at https://github.com/Xingyetan89/Data-processing-for-investigation-on-population-dynamics.git.

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

## Acknowledgements

We would like to express our gratitude to Dr. Lei Dong for his assistance in acquiring the Boston dataset. B.H. acknowledges support from the National Key Research and Development Program of China (Grant No. 2022YFB3903700), the National Natural Science Foundation of China (Grant No. 42271439), and the Hong Kong Research Grants Council (Grant Nos. SRFS2324-4H02, GRF 14616323, GRF 17617024, and TRS T22-606/23-R). Q.W. and W.L. acknowledge support from the U.S. National Science Foundation (NSF) under Grant Nos. 2125326, 2228533, 2402438, and the Northeastern University iSUPER Impact Engine. The funders had no role in the study design, data collection and analysis, decision to publish, or preparation of the manuscript.

## Author contributions

B.H. and X.T. conceived and designed the research. X.T. and W.L. compiled the data, and both X.T. and B.H. built the models. All authors performed analyses. X.T. and B.H. wrote the manuscript, while M.B., Y.Z., Q.W., W.L., and P.G. interpreted the findings and provided feedback on the manuscript drafts.

## Competing interests

The authors declare no competing interests.
