## [Peer Review file · Nature Communications]

The spatiotemporal scaling laws of urban population dynamics

Corresponding Author: Professor Bo Huang

Version 0:

Reviewer comments:

Reviewer #2

(Remarks to the Author)

The paper focuses on a specific yet interesting and relevant aspect of urban mobility: how population presence fluctuates. The authors show that a power-law relation holds between fluctuation level and the time window length of aggregation, as well as between fluctuation level and distance from city hubs (city center, main communication corridors and similar). The two dimensions (space and time) are found to be inter-dependent, one affecting the power-law exponent of the other. Also, relations against urban density are explored, considering population density or density of Points of Interest (POIs). Empirical results are based on mobile phone data coming from 4 Chinese cities, the large Boston area (USA) and an Italian city.

While I found the approach overall sound and the insights very interesting, there is a number of points that the authors might consider:

- Formula (1) and (to a minor extent) (2): differently from the authors, a first inspection of Figures 1d,e suggests to me quasi-constant-sloped curves with increasing offsets w.r.t. "r". The relation that would derive from that would look like $s^{\alpha(r)}$ ($c(\text{city})$ being a city-dependent constant) instead of Formula (1). A more detailed discussion of the Figure to defend the proposed model should be provided, since this is a critical choice.

- Fluctuations are computed as a standard deviation of (detrended) values. As such, it seems to me extremely natural to expect that fluctuations levels grow with population volumes -- and later experiments show that there is indeed a strong relation. This raises several questions that I think the authors should address:

- 1) what is the relation between fluctuation level and average value of each sub-sequence (of length s) generated? Notice that the experiments provided relate to real population from WorldPop or similar, not the original time series;
- 2) why not separating the two components -- time series level / mean value vs. deviations -- by analyzing a normalized version of standard deviation? E.g., by applying standard z-score normalization to each original sub-sequence. It would be interesting to see what is preserved by removing this strong (and slightly obvious) dependence from mean level of time series.
- 3) does all this provide some explanation for the increase of fluctuations w.r.t. the aggregation period "s"?

- The spatial aggregation is performed on constant-width rings with variable diameters. That leads to have more cells aggregated together the farther we go from the center (a linear growth, more exactly). What kind of impact does it have on the computed statistics? Also, the aggregation is such that the same cell might belong to more than one ring, contributing to different values of "r". Can this effect be considered negligible?

- The mathematical derivations provided in the Methods contain some unclear steps.

- 1) In the last step of Formulas (12) a term " $d_1 \ln r$ " appears after integration. Where does it come from?
- 2) According to (13), Formula (2) actually contains a multiplicative factor that depends on "s" (as it does the exponent of the polynomial), thus is not constant. Is (2) still a proper power-law?

- Extended Data Fig. 2 shows points corresponding to negative values of the observed $F(s,r)$. How is that possible?

- Suppl Note 3 describes a strategy for validating the stability of the results found relating fluctuations and distance from city

center. Yet, two issues emerge:

- 1) the description of the strategy is not sufficiently detailed, as it is not clear how rings are translated (by what amount, and how the randomization works);
- 2) the validation approach looks questionable to me: parameters' fitting and test/validation occur over the same cells, though aggregated in a different way. What is the robustness of this strategy? Some supporting discussion should be provided.

- The allometry model (from line 230) looks clear and interesting, yet the description and mathematical derivations leave some issue:

- 1) the discussion (lines 231-245) leaves unclear whether the population volume is measured empirically or if it is substituted by a standard model (exponential or polynomial). Later experiments make use of data, thus clarifying the point a bit, but the discussion of the model should be more direct about that;
- 2) the inverse power model is chosen as preferable, yet it is not clear if that plays any role in the discussion that follows in that section. What is the use of it?
- 3) Formula (6) is provided, yet I could not understand if that is an assumption, an empirical observation or a result of a mathematical derivation -- in the latter case, where does it come from? Please, clarify this aspect.

- The exception of Shenzhen (which needs two models for different ranges of "r") was discussed but not completely justified, in my opinion. How does it fit the overall hypothesis? Under what conditions can we expect to have cities with a single-scaling process, and when cities with multiple scales? Also, Extended Data Fig. 3 show that something similar happens in Milan, to some extent. Are the two phenomena connected?

- The datasets adopted come from mobile phones, but through different applications and thus potentially describing different populations with their own biases. The paper should discuss this aspect, in order to convince that the datasets correctly capture population volumes (and thus fluctuations) and that they are coherent. For instance, Deville 2014 (<https://www.pnas.org/doi/10.1073/pnas.1408439111>) shows that while CDR data volumes can model well population, extra care should be taken, since the relation can be non-linear, in their case by over-estimating population in crowded areas. Could this affect the fitting of the proposed models for fluctuations?

- A final note on reproducibility: part of the results are based on POI data that apparently cannot be shared, thus making it impossible to other researchers to replicate the same experiments. If that is the case, I would suggest to try replacing those data with OSM data, as for the other two use cases. Unless OSM is unusable for the task, in which case some justification should be provided.

Minor comments:

- Line 46: is "orthogonal" the right term? It suggests complete independence, whereas fluctuations are most likely (loosely) connected or to some extent derived from flows and trajectories.
- Figure S1: each map adopts different colors, and in several cases the contrast between mapped values and background is insufficient. I suggest to standardize all maps and choose an easier-to-read color selection. Also, the comparison between this and Figure 2 is not easy.
- Figures S8-12 contain several very small details -- especially numbered hotspots -- that are hardly visible (even on screen after zooming). Some improvement to readability is recommended.
- Line 190: the reference to Suppl Note 4 and Fig. S7 is unclear to me. I cannot see the connection with the current paragraph.
- Lines 318-319: "[This] provides a cogent response to the long-standing speculation that SOC with a signature of '1/f noise' could be related to the spatial structure of matter". I cannot see the relation between the results described in the previous paragraph and this statement.
- Line 525: what is the "Euclidean dimension of the embedding space"?

(Remarks on code availability)

The shared code provides the main functionalities discussed in the paper, yet it is quite minimal and I believe it should be improved in a few directions:

- the preprocessing functions are given only for Tencent's datasets, and it is very specific, including fixed thresholds, file names, etc. Similar functions should be provided for the other datasets, starting from the data format adopted by the data providers;
- the code for TWDFFA and predictions does not provide a master function to directly replicate the battery of experiments described in the paper, leaving the reader to do that. The code only applies a fixed configuration to few toy sample preprocessed datasets included directly in the github/codeocean page;
- the code is in general quite "barebone" and completely uncommented, making it difficult to reuse it if any modification is needed to replicate the experiments and/or play with parameters. Although the source code is rather simple and short, I would recommend to improve its readability and help the reader willing to put their hands in it.

Reviewer #3

(Remarks to the Author)

This research aims to reveal the underlying spatiotemporal scaling laws of urban population dynamics. Using large-scale mobile phone tracking data aggregated at square grids, the research found that the population time-series fluctuations at any given location adhere to a time-based scaling law with a spatially decaying exponent, which offers a quantifiable relationship between urban form and population dynamics. It is a solid research to advance our understanding of the scaling law of urban human mobility not only in spatial scale but also in temporal scale. I support its publication with addressing the following issues in the revision:

(1) City center definition and hierarchy impact: Are you using the population weighted center or geometric center of a city? Do you only consider one major center in each city for the distance to center scaling analysis? As described in the Supplementary Note 2 (Urban spatial structure and centrality), there exist multiple levels of "centers" in sub-regions of each city. Whether the scaling patterns can hold or not when using multiple centers for analysis in each city. This is very important for multi-core metropolitan areas which are known as polycentric cities in urban morphology.

(2) Urban form refers to the size, shape, and configuration of an urban area. There are different elements of urban forms in the literature. It seems to me that the authors mainly discussed the linkage to population density, POI density. The linkage to other elements needs to be discussed if keeping the term "urban form".

(3) Figure 1d: how are the distance bins (20, 60, 80km) chosen? Any relation to the urban area size of a city?

(4) Figure 2: It is very interesting to see the temporal decay patterns. However, how the daily rhythm of population fluctuations can be reflected in the temporal scaling patterns? How to link the scaling exponents to the morning/evening peak hours of urban population movements?

(5) Equation for the derivation of spatiotemporal scaling laws: what is d_1 in equation (12)? Please also clarify are you doing the Integral of $\ln s$ or $\ln r$ from the third line to the fourth line in equation (12).

(6) Method: Temporal scaling analysis: There are many time-series analysis methods, why did the authors choose the detrended fluctuation analysis (DFA) method to derive the relationship between the population fluctuation $F(s)$ and the temporal scale s .

(Remarks on code availability)

Since the authors didn't provide the original data, I cannot reproduce the results. However, based on the readme file information, sample data and the Matlab code, I think the results can be replicated to other studies by following the methods.

Version 1:

Reviewer comments:

Reviewer #2

(Remarks to the Author)

The authors took very seriously the many requests and questions I made in my original review, providing a detailed rebuttal that dealt systematically with each point.

Many issues I arose were due to a misunderstanding of the text, either due to text ambiguity (in which case the authors amended it by adding explicit remarks or fixing the terminology) or to a plain mistake on my side (in which case the authors explained it in clear terms).

The clarity issues in mathematical formulations and derivations have been fixed, mainly by either adding details to derivations or commenting them. The discussion seems now to me reasonably fluent and accessible.

In response to my issues on reproducibility of results, the authors also provided (in rebuttal letter) results of additional experiments on open data, to show their strong limitations but also to prove that they still fit the overall results obtained in the paper.

Details on the method and experiments were also added to the text.

Overall, I believe the authors did a great job in answering to my concerns, providing integrations where needed, and explaining/defending their work in a convincing way when all the answers were already there in the paper.

Only two small points were left behind:

1. Comment 3, about the effects of aggregating rings of increasing radius, was slightly misinterpreted: while the values of F are correctly weighted considering the overall area covered, larger rings will simply aggregate more data. The question is mainly a vague doubt: could this have any effect on statistic aggregations, like reducing their variance or similar?

2. Comment 4, part 2: I really expressed myself in the wrong way, apologies for that. My real aim was this: after stating (correctly) that (2) is a power law, we kind of forget the presence of a constant term containing "s", which might potentially be relevant in other derivations in the paper. Thus, I wanted to suggest to doublecheck that nowhere that really happened.

Both of them look minor to me, and I believe the authors can handle them without the need of another rebuttal cycle.

(Remarks on code availability)

The code provided covers now all phases of the experiments discussed in the paper. Answering to the requests of reviewers, the code seems now complete, better structured and also (acceptably) commented. Some data sources used in the paper are not open, so nothing can be done to make experiment completely reproducible, but the code itself is there, and in a usable form to be applied/adapted to similar experiments.

Reviewer #3

(Remarks to the Author)

The authors have done a great job to address all my comments and concerns. The paper has been significantly improved regarding its contribution, the clarity of research and equations to derive the spatiotemporal scaling law. It is a solid research to advance our understanding of the scaling law of urban human mobility not only in spatial scale but also in temporal scale.

(Remarks on code availability)

The reorganized code is clear and easy to follow.

RESPONSE TO REVIEWER COMMENTS

Reviewer #2:

Remarks to the Author

The paper focuses on a specific yet interesting and relevant aspect of urban mobility: how population presence fluctuates. The authors show that a power-law relation holds between fluctuation level and the time window length of aggregation, as well as between fluctuation level and distance from city hubs (city center, main communication corridors and similar). The two dimensions (space and time) are found to be inter-dependent, one affecting the power-law exponent of the other. Also, relations against urban density are explored, considering population density or density of Points of Interest (POIs). Empirical results are based on mobile phone data coming from 4 Chinese cities, the large Boston area (USA) and an Italian city. While I found the approach overall sound and the insights very interesting, there is a number of points that the authors might consider:

Response: We sincerely appreciate your positive feedback and thoughtful suggestions regarding our manuscript. Your comments have provided us with valuable insights, enabling us to gain a clearer and more comprehensive understanding of our study, both in theoretical and methodological terms. We are deeply grateful for your constructive input.

In response, we have carefully studied and reflected on each of your comments, and we have provided detailed, point-by-point responses below. Your inquiries and perspectives have not only helped us enhance the discussion of our research but have also allowed us to improve the structure and completeness of the manuscript.

Comment 1: Formula (1) and (to a minor extent) (2): differently from the authors, a first inspection of Figures 1d,e suggests to me quasi-constant-sloped curves with increasing offsets w.r.t. "r". The relation that would derive from that would look like $\alpha(r) s^{c(\text{city})}$ ($c(\text{city})$ being a city-dependent constant) instead of Formula (1). A more detailed discussion of the Figure to defend the proposed model should be provided, since this is a critical choice.

Response: We thank the reviewer for proposing such a constructive idea, which has allowed us to re-examine the underlying assumption and physical meaning of our proposed

model. The curves in Figs. 1d and 1e can be expressed as the following equations, (r1) and (r2), respectively, according to the mathematical model proposed in the comment and the one we established in this paper.

$$F(s, r) \propto \alpha(r)s^{c_i}$$

$$\text{Ln } F(s, r) = c_i \text{Ln } s + C_1(r), \quad (\text{r1})$$

where $C_1(r) = \ln \alpha(r)$, representing the intercept as a function of distance r ; and

$$F(s, r) \propto s^{\alpha(r)}, (\alpha(r) > 0)$$

$$\text{Ln } F(s, r) = \alpha(r) \text{Ln } s + C_2(r), \quad (\text{r2})$$

where $C_2(r)$ refers to the formula that characterizes the intercept as a function of distance r .

The essential difference between the two equations lies in whether the slope of the curve changes, which represents different assumptions for building models. Specifically, the underlying assumption of Eq. (r1) is that urban population fluctuations are governed by a unified scaling exponent and possess no spatial variation in a city, while Eq. (r2) assumes that the power exponent of these fluctuations varies over locations within a city.

Our analytical results in the manuscript have demonstrated that at the macro (city-wide) level, as shown in Figs. 1d and 1e, there are differences in the estimated slopes of these curves as listed in Supplementary Tables S4 and S5, characterized by Eqs. (4) and (5). At the micro (grid-cell) level, as shown in Fig. 2 and listed in Table 1, the more apparent slope differences between the curves and the reflected urban spatial structure indicate the spatial variation and heterogeneity of urban population fluctuations, which is consistent with the assumption suggested by Eq. (r2). Therefore, we think that our proposed model is more rational as it maintains the consistency of assumption between macro and micro levels and is thus more universal. To emphasize the heterogeneity of scaling exponents, we have supplemented the discussions with the following statement:

“Furthermore, the differences in these linear slopes indicate that the scaling exponents $\alpha(r)$ and $d(s)$ vary with spatial and temporal scales r and s , respectively.”

and

“Fig. 2 and Table 1 suggest that the estimated scaling exponents α_i vary across locations within individual cities. This confirms the spatial heterogeneity of temporal scaling processes in population fluctuations, as similarly revealed by Eq. (1).”

These additions were made in the first and second sections of the Results. Later, in the third section of the Results, Fig. 3, together with Extended Data Fig. 1, suggests the variation of scaling exponents over scales in a spatiotemporal gradient manner, validating the underlying assumption of our model and completing the expression of its parameters.

Comment 2: Fluctuations are computed as a standard deviation of (detrended) values. As such, it seems to me extremely natural to expected that fluctuations levels grow with population volumes -- and later experiments show that there is indeed a strong relation. This raises several questions that I think the authors should address:

1) what is the relation between fluctuation level and average value of each sub-sequence (of length $\$s\$$) generated? Notice that the experiments provided relate to real population from WorldPop or similar, not the original time series;

Response: We deeply appreciate your valuable inquiry, which prompted us to reflect on the methodology. From both theoretical and practical perspectives, we believe that there is no significant correlation between the fluctuation level and the average value of each sub-sequence for a single location.

Theoretically, detrended fluctuation analysis applies a detrending function to each sub-sequence, thereby removing the effect of average values (Kantelhardt *et al.*, 2001). Practically, population mobility within a specific time period (especially in the short term) is influenced not only by the population size but also by various factors, such as human travel behaviors, travel choices, and urban functional layouts. For instance, in a residential area, it is expected that the population would peak but mobility would be minimal at midnight, whereas during the daytime (particularly during rush hours), the population may decrease while mobility increases, resulting in more pronounced fluctuations. This indicates that there is no direct relationship between short-term fluctuation levels and population size for a single location. However, when viewed from a city-wide perspective across locations, a larger population may generally generate more mobility, which in turn leads to greater fluctuations.

To address this issue, we selected 9 samples with diverse population volumes and α_i values. Fig. R₁ demonstrates that when the time interval is small, global fluctuation trends result in more drastic variations in the average values of sub-sequences, with smaller fluctuations within the intervals. As the time interval increases, the average values of sub-sequences gradually converge because the increase in temporal scale reduces the number of sub-sequences, and fluctuations in population volumes are eliminated by the larger time scales. Despite this, the fluctuation level still increases, suggesting that the increase is caused by the larger time scales rather than changes in average values.

Comparing the cases in Fig. R₁, we observe that as the magnitude of the average values increases, the magnitude of the fluctuations also increases. This suggests that, in general, larger populations tend to cause greater fluctuations.

Fig. R₁ The plots of fluctuation level versus the average value of each sub-sequence for 9 selected samples.

2) why not separating the two components -- time series level / mean value vs. deviations -- by analyzing a normalized version of standard deviation? E.g., by applying standard z-score normalization to each original sub-sequence. It would be interesting to see what is preserved by removing this strong (and slightly obvious) dependence from mean level of time series.

Response: We sincerely thank the reviewer for their thoughtful suggestion to improve the research methodology. As explained above, there is no significant correlation between the mean value and the fluctuation level, so the mean value was not included as a factor for investigation. Additionally, the mean is scale-dependent; that is, the average values of sub-sequences are determined by the temporal scale used to divide them. This makes it impossible to fully separate the scale component from the mean component.

In this study, our primary focus is on the relationship between inherent fluctuations in population caused by human mobility and time scales. The root-mean-square displacement of the detrended profile directly quantifies the magnitude of these fluctuations without requiring normalization. Applying standard z-score normalization to each original sub-sequence, as suggested, could potentially distort and obscure important information about the fluctuations. For instance, such normalization could mask critical phenomena such as self-organized criticality (indicated by $1/f$ noise) and long-range correlations, which are key to understanding the underlying dynamics.

For these reasons, we believe that the use of z-score normalization would not yield true and accurate measurements and might hinder the ability to extract meaningful insights from the analysis.

3) does all this provide some explanation for the increase of fluctuations w.r.t. the aggregation period "s"?

Response: Once again, we would like to thank the reviewer for their thoughtful methodological advice. We believe that the observed increase in fluctuations with the aggregation period "s," which follows a scaling law, is due to the inherent long-range correlations present in complex systems and phenomena. This implies that fluctuations across different time scales are persistent, exhibiting similar increasing or decreasing trends over time. In such cases, larger aggregation periods capture more of this persistence, resulting in increased fluctuations.

Höll and Kantz (2015) demonstrated the derived relationship between the detrended fluctuation function $F(s)$ and the autocorrelation C_l , expressed as

$$F^2(s) = \langle x^2 \rangle \left(W(s) + \sum_{l=1}^{s-1} C_l L_l(s) \right) \quad (\text{r3})$$

where $W(s)$ and $L_l(s)$ are the deterministic functions dependent on the detrending order q , lag l , and scale s . For time series with long-range correlation, the autocorrelation function follows a power law, $C_l \sim l^{-\mu}$, with $\mu \in (0,1)$. From this, the detrended fluctuation function can be deduced as a power-law relation, $F(s) \sim s^\alpha$, where $\alpha = 1 - \mu/2$.

Comment 3: The spatial aggregation is performed on constant-width rings with variable diameters. That leads to have more cells aggregated together the farther we go from the center (a linear growth, more exactly). What kind of impact does it have on the computed statistics? Also, the aggregation is such that the same cell might belong to more than one ring, contributing to different values of "r". Can this effect be considered negligible?

Response: We thank the reviewer for raising these important concerns regarding the estimates of mean fluctuations based on the circular ring structure. Indeed, as the distance of the ring from the center increases, its area also increases, leading to the inclusion of more grid cells. Additionally, the construction of circular rings may result in certain grid cells being split and divided among different rings.

To address the effects of the increasing circular area and grid-cell segmentation, we adopted a weighted averaging scheme to estimate statistics, as shown in Eq. (r4), which was introduced in the ‘Temporal scaling analysis’ section of the Methods:

$$F(s, r) = \sum_{i=j \sim n} A_i F_i(s) / \sum_{i=j \sim n} A_i \quad (\text{r4})$$

In this equation, $F(s, r)$ refers to the mean population fluctuation at distance r from the city center during a given time scale s ; $F_j(s) \sim F_n(s)$ represent the population fluctuations of grid cells $j \sim n$ intersecting the annular region at distance r from the city center; and A_i ($i \in [j, n]$) is the area of the intersection between each grid cell and the annular region. By using this approach, the calculation of the statistic for a ring only considers the proportion of each grid cell's area that lies within it. This eliminates the effect

of variation in the number of aggregated cells caused by the changing ring area. Furthermore, segmented cells contribute to the rings they belong to in proportion to their area within each ring, minimizing and balancing the impact of segmentation.

This approach, based on the concentric ring structure, has been applied in other research, such as the estimation of spectral flows in human mobility (Schläpfer *et al.* 2021), as well as in studies of population and urban land density (Chen 2010; Jiao 2015).

To validate whether grid-cell segmentation affects our model, we designed an additional sensitivity analysis (i.e., the original ‘Validation’ section in Supplementary Note 3). Details of this experiment can be found in our response to Comment 6. To address your concern, we have supplemented the content in the Methods section to clarify this issue:

“This scheme accounts for the potential segmentation of grid cells by the ring structure by calculating the proportion of each grid cell's area that overlaps with each annular region as a weighting factor. To assess the influence of such segmentation, we conducted a sensitivity analysis by altering the configuration of the annular structure and examining its effect on the results (Supplementary Note 3).”

Comment 4: The mathematical derivations provided in the Methods contain some unclear steps.

1) In the last step of Formulas (12) a term " $d_1 \ln r$ " appears after integration. Where does it come from?

Response: We thank the reviewer for pointing out the ambiguity regarding the derivation of Eq. (12). Below, we provide a more detailed derivation process to clarify this step:

$$\begin{aligned} \frac{d \ln F(s, r)}{d \ln s} &= \alpha(r) = -b \ln r + \alpha_1 \\ d \ln F(s, r) &= -b \ln r d \ln s + \alpha_1 d \ln s \\ \int d \ln F(s, r) &= \int -b \ln r d \ln s + \int \alpha_1 d \ln s \\ \int d \ln F(s, r) &= -b \ln r \int d \ln s + \alpha_1 \int d \ln s \\ \ln F(s, r) + C_1 &= -b \ln r (\ln s + C_2) + \alpha_1 (\ln s + C_3) \end{aligned}$$

$$\ln F(s, r) = -b \ln r \ln s - bC_2 \ln r + \alpha_1 \ln s + \alpha_1 C_3 - C_1 \quad (\text{r5})$$

where Eq. (r5) can be reorganized into Eq. (12) with $d_1 = -bC_2$ and $Const = \alpha_1 C_3 - C_1$. Thus, the final form matches Eq. (12) in the manuscript. We have supplemented the detailed derivation process based on Eq. (r5) in the revised manuscript.

2) According to (13), Formula (2) actually contains a multiplicative factor that depends on "s" (as it does the exponent of the polynomial), thus is not constant. Is (2) still a proper power-law?

Response: We sincerely thank the reviewer for raising this insightful question. Let us first focus on Eq. (12) as it serves as the original function for Eqs. (1) and (2). When considering the joint influence of spatial and temporal factors (e.g., the variables r and s in this case), Eq. (12) does not provide the logarithmic transformation of a standard power-law distribution. However, if one variable (e.g., s) is controlled while the other (e.g., r) is interpreted, Eq. (12) can be reorganized into Eq. (13) and subsequently transformed into a power-law function, as shown in Eq. (2). This transformation facilitates the understanding of the spatiotemporal processes underlying population fluctuations.

Eqs. (1) and (2), as illustrated in Figs. 1d and 1e, demonstrate that at a specific spatial location or temporal scale, urban population fluctuations exhibit a scaling law. However, this scaling phenomenon varies with changes in spatial location or temporal scale. To aid in understanding, we can draw an analogy with another phenomenon, such as the distance decay of urban density. Assume that the urban density distribution follows a power-law function expressed as Eq. (15):

$$\rho(r) = \rho_1 r^\beta, (\beta < 0).$$

In this case, the scaling exponent β may vary across cities, depending on their location. For a single city, its scaling exponent β may also vary over time (e.g., year by year), indicating dependence on temporal elements.

In our study, Eqs. (1) and (4), as well as Eqs. (2) and (5), quantitatively describe how the scaling process of population fluctuations within cities varies with spatial location and the measured time scale. This type of scaling behavior is one of the main contributions of our work. Relevant supporting statements can be found in Lines 222-228 (highlighted in yellow)

in the section titled ‘Spatiotemporal gradients of scaling exponents.’

Comment 5: Extended Data Fig. 2 shows points corresponding to negative values of the observed $F(s,r)$. How is that possible?

Response: We appreciate the reviewer’s careful attention to this issue in Extended Data Fig. 2 and Fig. S9. In both figures, we have used the logarithmic form of $F(s,r)$ to represent the predicted and observed values.

First, this approach is consistent with the basic equation expressed in Eq. (12), as well as its mathematical derivation and graphical representation, all of which are based on the logarithmic form. Additionally, as expressed in Eqs. (1) and (2), urban population fluctuations display dimensional inconsistencies across time and space, which make direct comparisons difficult. Using the logarithmic form addresses these inconsistencies, allowing all results to be plotted on a single graph for better comparability.

As a result of this transformation, both predicted and observed $F(s,r)$ values may appear negative when the original values are less than one. This does not imply negative fluctuations; rather, it reflects the logarithmic scaling of small magnitudes.

In the original title of Extended Data Fig. 2, we indicated that the values are expressed in a logarithmic manner. In the revised manuscript, we have supplemented the figure title with an additional explanation clarifying the reason for using the logarithmic transformation.

Comment 6: Suppl Note 3 describes a strategy for validating the stability of the results found relating fluctuations and distance from city center. Yet, two issues emerge:

1) the description of the strategy is not sufficiently detailed, as it is not clear how rings are translated (by what amount, and how the randomization works);

Response: We sincerely thank the reviewer for raising this important issue concerning validation, as it allows us to clarify both the methodology and its significance.

In our original approach, we used a MATLAB method to generate uniformly distributed pseudorandom integers to approximately determine two random translation distances for

each case. Specifically:

- For the five cases with a circular interval width of 3 km (Beijing, Shanghai, Guangzhou, Shenzhen, and Greater Boston), we generated two integers from 1 to 29 using this method and divided them by 10 to obtain the translation distances, resulting in an accuracy of 0.1 km. We consider this accuracy sufficient, as it is nearly 1/10 of the spatial resolution of the grid cells. For instance, in the Beijing case, we randomly obtained two integers, 25 and 28, which correspond to translation distances of 2.5 km and 2.8 km, respectively.
- For the Milan case, with a circular interval width of 0.5 km, we used an integer range of [1, 49], generating two random translation distances with an accuracy of 0.01 km.

However, this random approach may result in cases where two sets of translation distances are too close to each other, or where the translated circular structure is nearly identical to the original structure. To address this, the results initially presented in Supplementary Note 3 are based on two customized translation distances with specific intervals. Specifically:

- For the five cases (Beijing, Shanghai, Guangzhou, Shenzhen, and Greater Boston), the translational distances were set to 1 km and 2 km.
- For the Milan case, the translational distances were set to 0.25 km and 0.35 km.

In the revised Supplementary Note 3, we have also included the results using randomly generated translation distances, labeled by their corresponding values. Additionally, we have supplemented the description of the randomization method and clarified the translational accuracy.

2) the validation approach looks questionable to me: parameters' fitting and test/validation occur over the same cells, though aggregated in a different way. What is the robustness of this strategy? Some supporting discussion should be provided.

Response: We sincerely appreciate the reviewer's insightful question, which has helped us improve the discussion on validation. Due to the common issues of limited access to data capturing human dynamics and the lack of ground-truth estimates (Barlacchi *et al.*, 2015; Batista *et al.*, 2020; Dong *et al.*, 2024; Pappalardo *et al.*, 2023), validating the results using

other referenceable data for the same case is highly challenging.

However, as stated in our response to Comment 9, we utilized multi-source data from different countries and cities, each with varying population coverage and collection methods. Using these datasets, we initially discovered the regularities presented in the manuscript within Chinese cities and subsequently validated them using two cities from other continents. We designed the experiment described in Supplementary Note 3 for the following reasons:

a) Testing the impact of cell segmentation:

As mentioned in Comment 3, it is crucial to examine whether our results are influenced by grid-cell segmentation. Redesigning the circular structure leads to different segmentation, which allows us to investigate this issue.

b) Testing sensitivity to distance sampling:

The statistical results in the main manuscript are based on specific distance sampling. We needed to verify whether these results would vary with different sampling approaches.

c) Testing parameter robustness:

Considering that the calibration of model parameters was conducted based on a specific sample, we aimed to evaluate whether these parameters could predict the values for other samples.

In this study, such an experiment was conducted on the same cells primarily due to constraints related to data format and the characteristics of population mobility. Specifically:

- For the four Chinese cities and Milan, we used check-in data and gridded call detail records, respectively. The grid cells employed were at the minimum spatial resolution for these cases (the Milan grid was predefined by the data provider). Reprojecting these data at the same resolution was nearly impossible.
- Lowering the spatial resolution to enable reprojection by aggregating grids would lead to the Modifiable Areal Unit Problem (MAUP). In our case, this would mean that mobility between the original grids within the newly aggregated grid would be

ignored, thereby affecting the estimation of population fluctuations and model parameters.

Nevertheless, we believe that the spatiotemporal scaling regularities we have discovered will remain valid even if the grid resolution changes. This belief is supported by the fact that the grids used for the Chinese cities, Milan, and Greater Boston differ in spatial resolution, yet the results remain consistent.

In the revised manuscript, considering that the main purpose of this section is to examine the sensitivity of the analytical results to the experimental design, we have renamed Supplementary Note 3 as ‘Sensitivity analysis’ to clarify its function and avoid any misunderstandings. We have also added further explanations on the issues and rationale behind the experiment design of sensitivity test. Please refer to the revised first paragraph in Supplementary Note 3 (highlighted in yellow) for additional details.

Comment 7: The allometry model (from line 230) looks clear and interesting, yet the description and mathematical derivations leave some issue:

1) the discussion (lines 231-245) leaves unclear whether the population volume is measured empirically or if it is substituted by a standard model (exponential or polynomial). Later experiments make use of data, thus clarifying the point a bit, but the discussion of the model should be more direct about that;

Response:

We thank the reviewer for raising this important issue regarding clarity. As described in the section “A new urban allometry of population dynamics,” two conventional models are commonly used to characterize the distance-decay distribution of urban density from the center to the periphery: the negative exponential model and the inverse power model. Considering that the inverse power model provides a more rational interpretation of population distribution based on its physical meaning, we selected it to characterize the spatial distribution of urban density.

The feasibility of this model selection was validated through the double-logarithmic relationships (as shown in Extended Data Fig. 3) derived from empirical measurements using population mapping data (WorldPop) and POI data (OpenStreetMap and Amap). In

general, the curves—especially those representing POI density—in Extended Data Fig. 3 exhibit sound linearity within certain ranges, supporting the suitability of the inverse power model. However, for the population case of Milan, the performance of both models is less satisfactory due to the weak population density gradient around its center. In this case, urban functional attractiveness, as indicated by POI density, may provide a more effective explanation for the spatial gradient pattern of population fluctuations.

For the revised manuscript, we have made the following clarifications:

a) The relationship between the models and the allometric scaling:

We clearly stated that the inverse power-law model and the spatiotemporal scaling model can be combined to yield an allometric scaling model. This is reflected in the added statement:

“In line with this notion, the inverse power model of urban density and the spatiotemporal scaling law of population fluctuations yield a spatiotemporal allometric relationship through a simple derivation [Eq. (16)]”

b) Validation using empirical data:

We explicitly clarified that the derived allometry was validated using empirical data and presented the results in Fig. 4. This is reflected in the revised statement:

“We validated the derived allometry in Eq. (6) using empirical data on population mapping and POI (Methods).”

These revisions aim to improve the clarity of the discussion and ensure that the derivation and validation process are more transparent to readers.

2) the inverse power model is chosen as preferable, yet it is not clear if that plays any role in the discussion that follows in that section. What is the use of it?

Response: We thank the reviewer for highlighting the ambiguity in this section. Our goal here is to investigate how population density and urban functional attractiveness, as the main factors influencing human mobility, correlate with urban population fluctuations. In this context, we use the density of POIs to characterize urban attractiveness—where denser

POIs indicate greater attractiveness. Determining the appropriate functional form to characterize urban density distribution is essential for establishing and verifying the mathematical relationship between these factors and population fluctuations.

Specifically, the inverse power model plays the following roles:

a) Derivation of the allometric model:

By using the inverse power model in Eq. (15), we derive the allometric model in Eq. (6), which reveals the correlations between population fluctuations and the two factors (population density and urban attractiveness). Empirical data is then used to test the validity of this derived model. Details about the derivation and verification are provided in our next response.

b) Relationship between scaling exponent and urban density:

By combining the inverse power model in Eq. (15) with Eq. (4), we derive Eq. (18) (see “Derivation for the logarithmic density gradient of $\alpha(r)$ ” in Methods). This equation demonstrates how temporal scaling exponent $\alpha(r)$ is related to urban density $\rho(r)$, showing how population density and urban functional attractiveness influence the long-term correlation of population fluctuations. This relationship helps explain the two distinct dynamic processes of fractional Gaussian noise (fGn) and fractional Brownian motion (fBm).

The inverse power model thus provides critical insights into how these factors affect urban self-organized criticality from the perspective of human mobility. For example, $\alpha(r) = 1$ serves as an indicator of such criticality.

3) Formula (6) is provided, yet I could not understand if that is an assumption, an empirical observation or a result of a mathematical derivation -- in the latter case, where does it come from? Please, clarify this aspect.

Response: We sincerely thank the reviewer for raising this important question regarding the ambiguity of Eq. (6). As we previously explained, Eq. (6) is the result of a mathematical derivation, which we later validated using empirical data to confirm the allometric relationship it represents.

Specifically, we derived Eq. (6) by combining two power-law relationships that describe spatial distance r as a variable from the center to the periphery:

- The spatial scaling law of urban population fluctuations, expressed in Eq. (2).
- The urban density model, represented by Eq. (15).

Based on Eq. (15), we derive the following transformation relationships:

$$r^\beta = \frac{\rho(r)}{\rho_1}$$

$$r^{d(s)} = (r^\beta)^{\frac{d(s)}{\beta}} = \left(\frac{\rho(r)}{\rho_1}\right)^{\frac{d(s)}{\beta}} \quad (\text{r6})$$

where ρ_1 , β , and $d(s)$ are constants for a given time scale s . Substituting Eq. (r6) into Eq. (2), we obtain Eq. (6) as follows:

$$F(s, r) \propto r^{d(s)} \propto \left(\frac{\rho(r)}{\rho_1}\right)^{\frac{d(s)}{\beta}} \propto \rho(r)^{\frac{d(s)}{\beta}} \propto \rho(r)^{A(s)} \quad (\text{r7})$$

The power-law relationship described in Eq. (6) indicates a linear relationship after applying a double-logarithmic transformation. This relationship was validated using empirical data, as shown in Fig. 4 (also in Fig. S10, 11) with the estimation results listed in Extended Data Table 3 and Supplementary Table S2. To clarify the content and improve transparency, we have made the following modifications:

- Clarified the derivation of Eq. (6):

We have added a detailed derivation process for Eq. (6) in the newly introduced section “Derivation of urban allometry” in Methods. This section also links to Eq. (16) for additional context.

- Referenced prior responses:

We have referred to and aligned this explanation with our prior response to ensure consistency in addressing the reviewer’s concerns.

These changes aim to enhance the clarity of the discussion and make the derivation process of Eq. (6) more explicit for readers.

Comment 8: The exception of Shenzhen (which needs two models for different ranges of "r") was discussed but not completely justified, in my opinion. How does it fit the overall

hypothesis? Under what conditions can we expect to have cities with a single-scaling process, and when cities with multiple scales? Also, Extended Data Fig. 3 show that something similar happens in Milan, to some extent. Are the two phenomena connected?

Response: We thank the reviewer for raising this thoughtful question about multi-scaling processes. Urban spatial growth is inherently a multiscale organizational process (Batty and Longley, 1994; Frankhauser, 1994) and is often accompanied by multifractal features (Murcio *et al.*, 2015). This implies that the spatial evolution of a city may exhibit varying growth probabilities across different locations or scales, thereby resulting in multiple scaling processes (Chen and Wang, 2013). For example, there can be significant differences in growth probabilities between urban areas and rural areas. Urban areas tend to exhibit more clustered and dense organization, while rural areas are typically more dispersed and sparsely distributed, leading to distinct spatial scaling characteristics.

The approach of identifying crossover points to determine different scaling ranges has previously been used to define urban boundaries from a morphological perspective (e.g., Tannier *et al.*, 2011). These multiple scaling features are not limited to spatial dimensions but can also occur temporally, indicating periodicity (e.g., Ge and Leung, 2013).

In our case, patterns of human mobility are influenced by urban spatial organization, including population distribution and the spatial layout of infrastructure or public facilities. As a result, population fluctuations may exhibit more than one scaling process, as observed in the Shenzhen case (Fig. 1e). This phenomenon is consistent with the idea of urban multi-scaling organization over space, as shown in Extended Data Fig. 3. For each scaling range of population fluctuations in Shenzhen, we demonstrated that it aligns with our proposed model and conforms to our overall hypothesis.

However, in the case of Milan (as shown in Extended Data Fig. 3a), the situation differs from that of Shenzhen. Milan does not exhibit a clear range of linearity to indicate a distinct scaling process. As we explained, the spatial ranges near Milan's city center do not show significant distance decay in population density but do exhibit decay features in population fluctuations. This may be due to the dominant influence of urban functional attractiveness in these areas, rather than population density.

To supplement the discussion of the multi-scaling phenomenon in Shenzhen, we have

added the following explanation:

“This is a common phenomenon that reflects different growth probabilities across locations or scales, leading to distinct scaling properties within multi-scaling processes⁴⁰. Each scaling process, when considered separately, conforms to the scaling law described in Eq. (2).”

Comment 9: The datasets adopted come from mobile phones, but through different applications and thus potentially describing different populations with their own biases. The paper should discuss this aspect, in order to convince that the datasets correctly capture population volumes (and thus fluctuations) and that they are coherent. For instance, Deville 2014 (<https://www.pnas.org/doi/10.1073/pnas.1408439111>) shows that while CDR data volumes can model well population, extra care should be taken, since the relation can be non-linear, in their case by over-estimating population in crowded areas. Could this affect the fitting of the proposed models for fluctuations?

Response: We sincerely thank the reviewer for raising this important issue regarding the potential biases associated with mobile phone data in investigating human activities. We fully agree that estimating real population volumes and fluctuations using mobile phone data can introduce deviations due to factors such as incomplete group coverage, users' habits and usage frequency, and variations in data collection methods.

In the context of urban population fluctuations, we believe that these fluctuations primarily originate from the moving population (i.e., human mobility) rather than the static population volume, which typically reflects the stay population. Therefore, while mobile phone data may overestimate population volumes in crowded areas, as noted in Deville *et al.* (2014), this overestimation may not necessarily affect the modeling of population fluctuations. However, we acknowledge that the effectiveness of mobile phone data in estimating dynamic population flows at multiple time scales is not yet fully understood.

Biases in mobile phone data are a common issue, and there is currently no simple, standardized solution to address them, particularly in the absence of ground-truth estimates (Pappalardo *et al.*, 2023). Unlike population mapping within a specific time scale, which can be compared and validated against census or survey data, it is challenging to assess the potential biases of mobile phone data in capturing real population fluctuations. This is due

to the lack of alternative data sources that provide accurate or comparable information on population dynamics (Batista *et al.*, 2020; Barlacchi *et al.*, 2015). Nevertheless, mobile phone data is widely recommended as a universal proxy for representing the presence and movement of people (Dong *et al.*, 2024).

To mitigate the influence of data biases on our findings, we utilized multi-source mobile phone datasets collected from different regions, years, and population groups. These datasets include:

- Tencent data, covering approximately 70% of the population,
- Telecom Italia data, covering nearly 30% of the population, and
- SafeGraph data, covering around 15% of the population.

Despite the differences in group coverage, user habits (e.g., app usage or cultural differences), and data collection methods (e.g., GPS vs. antenna-based methods), as well as non-data factors such as city size or spatial structure, we obtained consistent conclusions across all datasets. This consistency suggests that our proposed model is largely unaffected by these biases.

It is worth noting that our aim is not to make precise estimates, as in population mapping or human mobility modeling, but rather to uncover the spatiotemporal regularities of population dynamics. Given this objective, we believe that the proposed model is both feasible and universal in characterizing real population fluctuations. To address this issue, we have added a new paragraph discussing these arguments in the Discussion section (highlighted in yellow in the revised manuscript).

Comment 10: A final note on reproducibility: part of the results are based on POI data that apparently cannot be shared, thus making it impossible to other researchers to replicate the same experiments. If that is the case, I would suggest to try replacing those data with OSM data, as for the other two use cases. Unless OSM is unusable for the task, in which case some justification should be provided.

Response: We greatly appreciate the reviewer’s valuable suggestion regarding data use for reproducibility. Initially, we intended to utilize the POI data from OpenStreetMap (OSM) for our analysis of Chinese cities. However, due to the impact of relevant policies and laws,

the use of OSM in mainland China is restricted. This has resulted in significant inaccuracies and missing data for many regions in mainland China.

To illustrate these issues, we obtained OSM data for China in 2018 from Geofabrik (<https://download.geofabrik.de/asia/china-180101-free.shp.zip>) and extracted its POI information for analysis. Unfortunately, we found that the OSM data for the studied cities is severely incomplete, especially for Guangzhou and Shenzhen. The total volume of POI data in OSM is significantly smaller compared to Amap. For instance, the OSM dataset for Shenzhen contains only around 1,300 records, whereas the POI data from Amap includes nearly 250,000 records. This large discrepancy is particularly evident in peripheral areas. For example, OSM contains only 21 POIs in areas located 30 km from Shenzhen's city center (which accounts for more than 30% of its administrative area), and the average number of POIs in circular areas spaced 3 km apart is only 2.

Despite this severe deficiency, we analyzed the allometric relationships between the mean population fluctuations $F(s, r)$ and POI densities $\rho(r)$ using OSM data for four Chinese cities. The results are shown in Fig. R2 and Table R1. Even with the inaccuracies and sparse data in peripheral areas, urban population fluctuations and POI densities still exhibit a certain degree of allometric relationship in Beijing, Shanghai, and Guangzhou. In contrast, Shenzhen shows a weaker relationship, primarily due to four outliers in the outermost periphery. After excluding these outliers, the relationship improves significantly, with the goodness of fit (R^2) ranging between 0.760 and 0.824, and an average R^2 of 0.800.

To address the reviewer's concerns, we have revised the content in the Methods section to clarify our approach:

“To maximize the accuracy of POI acquisition while ensuring data availability, the POI data for the city of Milan and the region of Greater Boston were directly downloaded from OpenStreetMap (<https://www.openstreetmap.org/>), while those for Chinese cities were collected from Amap (<https://www.amap.com/>) due to the significant lack of Chinese POI records in OpenStreetMap.”

Fig. R2 Urban allometric relationships between the mean population fluctuations $F(s, r)$ and POI densities $\rho(r)$ estimated using OSM data.

Table R1 Ranges and averages of the goodness of fit, R^2 for estimating allometric relationships between population fluctuations and POI densities using OSM data

City Name	R^2	Avg. R^2
Beijing	0.861 – 0.875	0.866
Shanghai	0.776 – 0.815	0.789
Guangzhou	0.692 – 0.726	0.704
Shenzhen	0.268 – 0.336	0.294

Minor comments:

Comment 11: Line 46: is "orthogonal" the right term? It suggests complete independence, whereas fluctuations are most likely (loosely) connected or to some extent derived from flows and trajectories.

Response: We thank the reviewer for highlighting the need for precision in wording. Conventionally, time is often regarded as a dimension independent of space. However, our

derived relationship, supported by empirical case studies, suggests that time and space exhibit some degree of correlation in characterizing population fluctuations. Therefore, as you correctly pointed out, the use of the term "orthogonal" is not accurate in this context.

As expressed in Eq. (12), the parameter b quantifies the combined impact of time and spatial scales on population fluctuations. Given the small values of the estimated parameter b (as listed in Extended Data Table 2), we have revised the corresponding sentence as follows:

“However, the aspect of population fluctuations driven by human mobility, which can generally be regarded as occurring on a dimension approximately perpendicular to the plane encompassing trajectories and flows, has never received adequate attention.”

Comment 12: Figure S1: each map adopts different colors, and in several cases the contrast between mapped values and background is insufficient. I suggest to standardize all maps and choose an easier-to-read color selection. Also, the comparison between this and Figure 2 is not easy.

Response: We greatly appreciate the reviewer’s constructive suggestion and for pointing out these important issues.

First, we have standardized the color grading of the maps depicting urban spatial structures, ensuring consistency with the subsequent maps illustrating urban hot spots. To improve readability, we also selected a more visually distinct and easier-to-read color palette to enhance the contrast between mapped values and the background.

Regarding the comparison with Fig. 2, we have reorganized the original Fig. S1 by splitting it into separate maps for each city. Additionally, we have attached the corresponding spatial patterns of temporal scaling exponents (α_i) from Fig. 2 for direct comparison, resulting in the current Figs. S1–S6 in Supplementary Note 2.

Comment 13: Figures S8-12 contain several very small details -- especially numbered hotspots -- that are hardly visible (even on screen after zooming). Some improvement to readability is recommended.

Response:

We thank the reviewer for highlighting the issue of unclear and unreadable maps. In response, we have remade the maps identifying urban hotspots, focusing on improving magnification and ensuring the clear presentation of legends, annotations, and labels. Additionally, we have rearranged these maps to enhance their clarity and readability. Please refer to Figs. S13–S18 for details on the modifications.

Comment 14: Line 190: the reference to Suppl Note 4 and Fig. S7 is unclear to me. I cannot see the connection with the current paragraph.

Response: We thank the reviewer for pointing out this issue. To address your concern, we have removed the reference to **Supplementary Note 4** and **Fig. S7** from this section.

Comment 15: Lines 318-319: "[This] provides a cogent response to the long-standing speculation that SOC with a signature of '1/f noise' could be related to the spatial structure of matter". I cannot see the relation between the results described in the previous paragraph and this statement.

Response: We appreciate the reviewer's valuable comment. It is widely recognized that urban systems, like other complex systems in physics, biology, and economics, exhibit self-organized criticality (SOC). SOC refers to a critical state of dynamical systems far from equilibrium, characterized by scale-free structures and the absence of characteristic temporal or spatial scales (Bak, Chen, and Creutz, 1989; Batty and Xie, 1999). Urban SOC is typically identified by three key features: fractals in space, rank-size distributions following Zipf's law, and $1/f$ noise in time (Chen and Zhou, 2006). While the first two features have been extensively studied and documented in urban systems, the existence of $1/f$ noise in urban phenomena remains less explored.

It is important to note that $1/f$ noise is more accurately expressed as $1/f^\gamma$ noise (Mandelbrot and Van Ness, 1968), with the spectral index $\gamma = 1$ being a special case. The most straightforward way to detect $1/f^\gamma$ noise is by using power spectral density (PSD) analysis, which reveals the scaling relationship between the power spectrum $S(f)$ and frequency f in time series. However, since PSD analysis is limited to stationary time series, we instead applied detrended fluctuation analysis (DFA), which can handle non-stationary time series. DFA reveals the scaling relationship between fluctuations and time

scales, and the scaling exponent α_i (or $\alpha(r)$) obtained from DFA can be converted to the spectral index γ from PSD, allowing DFA to indicate $1/f^\gamma$ noise. (Please refer to Supplementary Note 4 for details.)

In this study, we demonstrated not only the existence of $1/f$ noise in urban phenomena, as indicated by the scaling law between population fluctuations and time scales, but also its spatial correlation. Specifically, we found that high values of the temporal scaling exponent α_i are concentrated in city and town centers, major urban areas, and important transportation corridors, while low values are distributed in suburban or exurban areas. This forms a general gradient trend from the center to the periphery (as shown in Figs. 2 and 3 and statistically expressed by Eq. (4)) and varies with density (as shown in Extended Data Fig. 5 and statistically expressed by Eq. (18)). These findings demonstrate that the spectral index γ is also spatially correlated.

Thus, we concluded that urban SOC with a signature of $1/f$ noise is related to urban spatial structure. To clarify the indicative role of the temporal scaling exponent, we revised the statement in the section “Heterogeneity of temporal scaling and regularity of spatial organization” as follows:

“The exponent α_i also has a mutually transformable relationship with the Hurst exponent and the spectral index. Thus, it can characterize the long-range dependence of the time series⁴¹ and indicate the phenomenon of $1/f$ noise (Supplementary Note 4 and Table S3).”

Comment 16: Line 525: what is the "Euclidean dimension of the embedding space"?

Response: We thank the reviewer for this inquiry. The term "Euclidean dimension" refers to the number of independent coordinate axes typically used to describe or measure an object, and it is expressed as an integer. For example, a straight line can be represented using one-dimensional coordinates, while a plane requires two-dimensional coordinates.

When dealing with fractal structures that have non-integer dimensions, we use Euclidean space to visualize and measure them. For instance, in the case of the spatial distribution of population density, we use a two-dimensional Euclidean plane to display or measure it. Therefore, the Euclidean dimension of the embedding space in this context is 2.

Reviewer #2 (Remarks on code availability):

The shared code provides the main functionalities discussed in the paper, yet it is quite minimal and I believe it should be improved in a few directions:

Response: We sincerely thank the reviewer for carefully pointing out the issues regarding the availability and readability of the shared code. Based on your comments, we have made modifications and improvements to the code. Please find our point-by-point responses below for detailed explanations.

Comment 17: the preprocessing functions are given only for Tencent's datasets, and it is very specific, including fixed thresholds, file names, etc. Similar functions should be provided for the other datasets, starting from the data format adopted by the data providers;

Response:

We thank the reviewer for this valuable comment. In the original submission, we only shared the preprocessing code for the Tencent dataset due to the complex structural and formatting issues involved in handling its raw data. Below, we provide an explanation of the preprocessing steps for Tencent data, as well as additional details regarding the other datasets.

Tencent Dataset:

Tencent's original check-in data is organized and integrated by time spots, meaning that each crawled file contains all check-in records worldwide at a specific time point. Consequently, the data for a single day is composed of multiple files captured at numerous time spots. To obtain the georeferenced time series described in the Methods, the preprocessing involves the following steps:

Extraction: For each time point file, we extract check-in records corresponding to the scope of the study area.

Integration: These extracted records are integrated into a new format on a daily basis. Specifically, a single file is generated for each day, where each row contains the time spot, latitude and longitude, and the number of check-ins.

Construction: Georeferenced time series are then constructed by using geographical coordinates and time as rows and columns, with the corresponding check-in records

filling in the matrix.

Milan Dataset:

The original data for Milan, derived from telecommunication antennas, has already been processed by its provider (Barlacchi *et al.*, 2015). It is gridded and converted into check-in records every 10 minutes for each grid cell on a daily basis. Simple data arrangement is sufficient to transform this dataset into the georeferenced time series format, making the preprocessing less complex compared to Tencent data.

Greater Boston Dataset:

The raw dataset for Greater Boston contains mobility records with timestamps, longitude, and latitude. To preprocess this data, the following steps are undertaken:

Mapping: Mobility records are mapped to their corresponding grid cells at 20-minute intervals.

Counting: For each unique combination of time interval and grid cell, we count the number of occurrences.

Integration: The resulting counts are grouped by time intervals and grid cells, and the final processed data is integrated into georeferenced time series.

In response to the reviewer’s comment, we have uploaded additional preprocessing code for the Milan and Greater Boston datasets. These additional codes provide similar functionalities, starting from the data formats adopted by the respective data providers, ensuring consistency and usability across all datasets.

Comment 18: the code for TWDFFA and predictions does not provide a master function to directly replicate the battery of experiments described in the paper, leaving the reader to do that. The code only applies a fixed configuration to few toy sample preprocessed datasets included directly in the github/codeocean page;

Response:

We sincerely thank the reviewer for raising these important issues regarding the functionality and parameter configuration of the provided code. In response, we have made the following revisions to address these concerns:

Added interactive parameter settings:

We have incorporated interactive prompts into the code, allowing users to set variable parameters (e.g., file names or key parameters for estimation) during code execution without needing to manually modify the source code. This enhancement significantly improves the flexibility and ease of use for different datasets and configurations.

Refactored the code into modular functions:

To improve clarity and usability, we split the source code into a series of modular functions, each corresponding to a distinct functionality.

Constructed a main function:

We created a main function to call these modular functions in sequence, enabling users to execute the full process directly. This restructuring makes the code easier to interpret, replicate, and modify for future experiments.

These updates ensure that the code is more user-friendly and better supports replication of the experiments described in the paper, while also facilitating future extensions and customization.

Comment 19: the code is in general quite "barebone" and completely uncommented, making it difficult to reuse it if any modification is needed to replicate the experiments and/or play with parameters. Although the source code is rather simple and short, I would recommend to improve its readability and help the reader willing to put their hands in it.

Our response:

We thank the reviewer for this valuable comment regarding code readability. In response, we have added detailed comments throughout the code, explaining its functionality and individual statements. These comments are intended to facilitate easier interpretation, modification, and replication of the experiments, as per your suggestion.

For more details, please refer to the updated code available on the GitHub/CodeOcean repository.

Reviewer #3

Remarks to the Author:

This research aims to reveal the underlying spatiotemporal scaling laws of urban population dynamics. Using large-scale mobile phone tracking data aggregated at square grids, the research found that the population time-series fluctuations at any given location adhere to a time-based scaling law with a spatially decaying exponent, which offers a quantifiable relationship between urban form and population dynamics. It is a solid research to advance our understanding of the scaling law of urban human mobility not only in spatial scale but also in temporal scale. I support its publication with addressing the following issues in the revision:

Response: We sincerely thank you for your encouraging feedback and positive support for our study. In particular, we deeply appreciate your favorable recognition of our main contribution to advancing human mobility research through the exploration of spatiotemporal scaling laws. We are also immensely grateful for the constructive and insightful comments you provided. These comments not only prompted us to critically examine the methodology and framework of this study but also inspired future directions for related research. We carefully studied and thoughtfully considered each of your comments, and we have provided detailed point-by-point responses. Your feedback has significantly helped us further improve the completeness and quality of the manuscript.

Comment 1: City center definition and hierarchy impact: Are you using the population weighted center or geometric center of a city? Do you only consider one major center in each city for the distance to center scaling analysis? As described in the Supplementary Note 2 (Urban spatial structure and centrality), there exist multiple levels of "centers" in sub-regions of each city. Whether the scaling patterns can hold or not when using multiple centers for analysis in each city. This is very important for multi-core metropolitan areas which are known as polycentric cities in urban morphology.

Response: We greatly appreciate the reviewer's valuable inquiries and insightful perspective. We strongly agree with your viewpoint that cities can possess polycentric structures, which are highly significant in urban studies.

In this study, the city centers we referred to and used as spatial references for measuring and modeling $F(s, r)$ correspond to the main urban centers structurally defined in the master plans of the studied cities (as shown in Figs. S1–S6). These centers were further confirmed and spatially delineated based on the temporal scaling analysis of population fluctuations at the grid-cell level (as shown in Fig. 2). Generally, these centers are located near the geometric center of the central cluster delineated by the classified α_i values (please refer to Figs. S13–S18). Among the six study cases, Shenzhen is unique in having multiple main centers, as defined in its master plan and supported by our temporal scaling analysis (please refer to Figs. 2, S4, and S16). For Shenzhen, we considered all its main centers and defined r as the distance from the closest main center.

The spatial pattern of the calculated α_i values based on population fluctuations revealed the polycentricity of each city (as shown in Figs. S1–S6). Consequently, the concentric ring structures built upon these spatial patterns, along with our measurements, statistical results, and model construction, inherently account for the impacts of multi-level centers within cities. As shown in Figs. 1 and 3, some abnormal oscillations or deviations of scattered points relative to linearity occur, especially at larger distances from the main urban center. These deviations may result from urban sprawl, which introduces imbalances and heterogeneities in the development of peripheral areas (e.g., the construction of new towns and suburban hubs). Despite these disturbances, the spatial distribution of urban population fluctuations generally adheres to our proposed scaling law.

Through these case studies, our primary aim is to demonstrate that, although cities exhibit uneven spatial development and organization (as described by the sector model or the multiple nuclei model, as shown in Figs. 2, S1–S6, and S13–S18), we can still identify scaling laws that characterize the spatiotemporal patterns of population fluctuations from the main urban centers outward, similar to the concentric zone model proposed by Burgess.

Nevertheless, we recognize that investigating spatiotemporal scaling based on urban polycentricity, as you suggested, is an important and promising direction for future research. However, it introduces several intricate and challenging issues:

a) Limited distance samples for scaling analysis:

Using multi-level centers for measurement, especially with data at the current resolution, reduces the range of distance r values, leading to insufficient distance samples for examining spatial scaling behavior and scale invariance.

b) Weighting the impacts of multiple centers:

Determining how to weigh the influence of multiple centers, either at the same hierarchical level or across different levels, is highly challenging. This difficulty is compounded by the lack of sufficient knowledge about the development status of these centers. For example, low-level peripheral centers may not yet exhibit clear scaling patterns due to insufficient development over a short period. The spatial scaling formation of such centers often requires gradual evolution—from the establishment of initial structures (represented as a point with zero dimension on a two-dimensional plane), through spatial filling via urban construction, to the formation of a relatively mature area (a fractal structure with a dimension between 1 and 2).

c) Cumulative effects of multiple centers:

The influence of low-level centers must also account for the cumulative effects of the main center and other centers. Determining the weight of these influences is extremely complex and context-dependent.

d) Data limitations on human mobility:

One potential approach is to use human mobility data to determine the attractiveness of each center to surrounding areas. However, such data is difficult to obtain and is often affected by time scale considerations, adding further complexity.

These challenges can lead to an excessive reliance on prior knowledge and introduce complicated constraints and mechanisms into the model-building process. For these reasons, we regard this topic as an important direction for future research.

To address the reviewer's concerns in the revised manuscript, we have clearly stated the city centers used for constructing the annular structure of each city (please refer to the highlighted content in the title of Fig. 1) and reorganized the discussion section on urban spatial organization, emphasizing that our findings inherently consider urban polycentricity (please refer to the highlighted content in the fourth paragraph). Additionally, we have integrated the contents of Comments 1 and 4 and supplemented the discussion with future

research prospects (please refer to our response to Comment 4 and the highlighted content in the last paragraph)

Comment 2: Urban form refers to the size, shape, and configuration of an urban area. There are different elements of urban forms in the literature. It seems to me that the authors mainly discussed the linkage to population density, POI density. The linkage to other elements needs to be discussed if keeping the term "urban form."

Response: We sincerely thank the reviewer for highlighting the need for conceptual and wording rigor. Indeed, "urban form" is a broad concept encompassing various aspects, such as size, shape, configuration, pattern, and structure, which emerge from the spatial organization of multiple urban elements.

In our study, we focused on population fluctuations caused by human mobility. While we acknowledge that other urban elements, such as land use and transportation infrastructure, can also influence population movements, we selected two key elements—population density and POI density—as they are fundamental factors directly relevant to our modeling.

Population density: Conventional population-mobility models, such as the gravity model (Zipf, 1946) and the radiation model (Simini *et al.*, 2012), are built upon information about population distribution.

POI density: Recent studies, such as Schläpfer *et al.* (2021), have revealed and quantified the impact of urban functional attractiveness on mobility, which we characterized using POI density.

Other urban elements are often spatially correlated with these two factors (Quinn, 2013; Jiang *et al.*, 2015). By deriving an allometric relationship using these two urban densities (population density and POI density), we aimed to provide more intuitive and meaningful indicators to interpret the general patterns and characteristics of population fluctuations. This allometric relationship was empirically validated using data from WorldPop, OpenStreetMap (OSM), and Amap.

In our case, the statistical properties of population fluctuations are closely tied to urban spatial organization, reflecting a progression:

From fractional Gaussian noise to fractional Brownian motion,
From stationarity to non-stationarity,
From weak to strong long-range correlation,
From the periphery to the city center,
In alignment with urban density, from low to high.

When the population or urban function at a particular location in a city reaches a critical density threshold (e.g., for the four Chinese cities studied, this is approximately 5,000 people/km²), it leads to population fluctuations exceeding the $1/f$ noise and producing a transition in properties. This relationship also provides insights into the dynamic characteristics of population fluctuations using commonly available static indicators.

To eliminate ambiguity in expression, we replaced the term "urban form" in the original abstract with "urban structure," as the latter better aligns with the context of this study.

Comment 3: Figure 1d: how are the distance bins (20, 60, 80km) chosen? Any relation to the urban area size of a city?

Response: We thank the reviewer for raising this question, which has made us aware of the potential misunderstanding caused by the original figure. In Fig. 1d, each quasi-linear plot corresponds to a specific distance from the main urban center. For each city, the distances represented by these plots are equidistant, as listed in Supplementary Table S4. The purpose of setting these distance bins is to visualize the relationships between population fluctuations and time scales at different distances.

As you speculated, the determination of the distance bins (e.g., 20 km and 80 km) in Fig. 1d was based solely on the spatial range of the case cities (i.e., the maximum radius from the closest main urban center). This approach reflects the differences in the scale of urban administrative areas.

However, we acknowledge that this division may cause potential ambiguity, as you have pointed out. To address this concern, we have revised Fig. 1d by dividing the maximum radius (denoted as r_m) of each city into three equal parts, creating new distance bins for each case. This approach aligns with the concept of temporal trisection used in Fig. 1e.

Comment 4: Figure 2: It is very interesting to see the temporal decay patterns. However, how the daily rhythm of population fluctuations can be reflected in the temporal scaling patterns? How to link the scaling exponents to the morning/evening peak hours of urban population movements?

Response: We sincerely thank the reviewer for these insightful inquiries. This study focuses on the temporal scaling behavior of population fluctuations on a daily basis. While we examined multiscale population fluctuations in temporal increments of 20 minutes, ranging from one hour to one day, the estimation of scaling exponents $\alpha(r)$ and α_i is based on a specific scaling range (from one hour to one day in our case), rather than specific peak hours. As a result, the temporal scaling patterns reflect the scale invariance of population fluctuations within a day, rather than the morning or evening peak situations of urban population movements.

As shown in Figs. 1d and Fig. 2, this time scale invariance demonstrates that, despite the daily rhythm of human activities (e.g., peak hours in the morning and evening associated with sunrise and sunset), the mean population fluctuation within one day approximately follows a power-law growth with increasing time scales. The diversity in such growth, as characterized by the scaling exponents, reflects different properties of population variation over time due to human mobility, including stationarity, non-stationarity, and long-term correlation with varying levels of persistence (strong or weak). The corresponding spatial patterns help reveal the spatiotemporal laws of population dynamics, which in turn indicate urban spatial organization and self-organized criticality.

Under the current research paradigm based on urban complexity, we recognize that further work could explore the daily rhythm of population fluctuations in greater detail, but this approach faces several challenges and uncertainties:

Crossover points for scaling ranges:

Crossover points (Kantelhardt, 2009) could potentially be identified in the plots of Figs. 1d and Fig. 2 to distinguish different scaling processes within a day. These scaling ranges, delineated by crossover points, may reflect hidden intrinsic periodicities of human activities within a day, analogous to periodicity observed in natural phenomena (Eichner *et al.*, 2003; Zhou and Leung, 2010). However, given that the double-logarithmic

relationships we obtained (as shown in Figs. 1d and Fig. 2) already exhibit high linear goodness of fit (averages of $\alpha(r)$ and α_i are all greater than 0.985, as listed in Table 1 and Extended Data Table 1), it is uncertain whether statistically significant and practically meaningful crossover points can be reliably identified.

Challenges in identifying crossover points:

There is currently no unified method for identifying crossover points, especially when addressing issues such as intrinsic deviations at small time scales (s) and insufficient sample sizes at larger time scales after delineation. Additionally, identifying crossover points for very large datasets, such as those in our study, would be computationally expensive and practically limited.

Multifractal analysis of time series:

Multifractal analysis could provide another way to examine the daily rhythm of population movements by analyzing the resulting multifractal spectrum, which reflects the contribution of fluctuations at different time scales. However, comparing multifractal spectra across a large number of cases (e.g., thousands of grid cells in a city, as in our study) would be computationally unfeasible and methodologically complex.

To address the reviewer's comment, we have incorporated your suggestions (together with those in Comment 1) as potential directions for future research. We have supplemented the Discussion section with the following statement:

“For example, deeper investigation into city polycentricity and daily human activity rhythms may be needed to improve predictions of population fluctuations. However, our research still faces data-related and methodological challenges and uncertainties within the current paradigm of complexity. These include insufficient data resolution for thorough examination of local scaling, as well as issues of statistical significance, computational complexity, comparability, and practical interpretation when examining temporally multi-scaling processes.”

Comment 5: Equation for the derivation of spatiotemporal scaling laws: what is d_1 in equation (12)? Please also clarify are you doing the Integral of $\ln s$ or $\ln r$ from the third line to the fourth line in equation (12).

Response: We thank the reviewer for identifying the issue regarding the omitted details in the derivation of Eq. (12). We have supplemented these missing details in the revised manuscript. Since Reviewer #2 also raised a similar concern, we have rewritten the derivation (now labeled as Eq. (r5)) for your review, as follows:

$$\begin{aligned}
\frac{d \ln F(s, r)}{d \ln s} &= \alpha(r) = -b \ln r + \alpha_1 \\
d \ln F(s, r) &= -b \ln r d \ln s + \alpha_1 d \ln s \\
\int d \ln F(s, r) &= \int -b \ln r d \ln s + \int \alpha_1 d \ln s \\
\int d \ln F(s, r) &= -b \ln r \int d \ln s + \alpha_1 \int d \ln s \\
\ln F(s, r) + C_1 &= -b \ln r (\ln s + C_2) + \alpha_1 (\ln s + C_3) \\
\ln F(s, r) &= -b \ln r \ln s - b C_2 \ln r + \alpha_1 \ln s + \alpha_1 C_3 - C_1 \quad (r5)
\end{aligned}$$

where Equation (r5) can be reorganized into Equation (12) with $d_1 = -b C_2$ and $Const = \alpha_1 C_3 - C_1$.

To clarify your specific question, the integration in the derivation is performed with respect to, $\ln s$ not $\ln r$, as indicated in the third and fourth lines of the derivation. We hope this explanation resolves the ambiguity and provides a clear understanding of the derivation process.

Comment 6: Method: Temporal scaling analysis: There are many time-series analysis methods, why did the authors choose the detrended fluctuation analysis (DFA) method to derive the relationship between the population fluctuation $F(s)$ and the temporal scale s .

Response: We greatly appreciate the reviewer's inquiry regarding our choice of method. The selection of methods is driven by our research objectives, and there are multiple reasons for using detrended fluctuation analysis (DFA) to study the temporal scaling of urban population fluctuations.

From the perspective of population dynamics, numerous studies have focused on human movements over space and uncovered spatially scale-free features of human mobility (Arcaute, 2020). However, population volume fluctuations over time at specific locations—another intuitive representation of population dynamics caused by human

mobility—have received relatively little attention. Building on existing findings on human mobility, we naturally raised the question of whether population fluctuations also exhibit scale invariance in the time domain, as seen in other natural phenomena (Goldberger *et al.*, 2002; Weber and Talkner, 2001; Kantelhardt *et al.*, 2003). If such temporal scale invariance exists, we further sought to explore its practical implications and underlying causes.

Additionally, population dynamics have traditionally been investigated and modeled at specific time scales. However, the results of such studies can vary depending on the statistical aggregates used within different temporal scales, similar to the modifiable areal unit problem in spatial analysis (Rozenfeld *et al.*, 2008). This highlights the need for a methodology that addresses the temporal scaling issue in population dynamics. Understanding this scaling behavior would allow us to extend the modeling and prediction of human mobility from a single time scale to multiple time scales, improving the robustness and generalizability of such models.

From the perspective of urban complexity, self-organized criticality (SOC) typically manifests in three forms: fractals in space, rank-size distributions following Zipf's law in hierarchy, and $1/f$ noise in time (Chen and Zhou, 2008). While the first two manifestations have been extensively studied in urban research, the third— $1/f$ noise in the time domain—remains largely unexplored. Investigating the presence of $1/f$ noise in cities would fill this research gap, advance the theoretical understanding of complex urban systems, and provide insights into the characteristics of urban SOC.

To investigate $1/f$ noise, the most straightforward method is power spectral density (PSD) analysis, as illustrated in Supplementary Note 4 (Mandelbrot and Van Ness, 1968). However, PSD analysis requires the use of fast Fourier transform (FFT) and is thus limited to stationary time series (DePetrillo and Ruttimann, 1999). While there are alternative methods to reveal the fractal nature of time-varying processes—such as rescaled range analysis (Hurst, 1951), scaled windowed variance analysis (Cannon *et al.*, 1997), dispersional analysis (Bassingthwaighte, 1988), and the average wavelet coefficient method (Simonsen *et al.*, 1998)—we selected DFA for several key reasons:

Broad applicability:

DFA can handle both stationary and non-stationary time series, making it well-suited for analyzing urban population fluctuations, which may exhibit non-stationary

characteristics.

Straightforward implementation:

Unlike other methods that require complex transformations, DFA is simple to implement and interpret, allowing for clear insights into the dynamics of time-varying processes.

Alignment with research objectives:

DFA succinctly characterizes the dynamics and long-term correlations of time series, aligning with our goal of studying temporal scaling effects in population dynamics.

Connection to $1/f$ noise:

The scaling exponent derived from DFA (expressed as $\alpha(r)$ and α_i in our study) has mutual conversion relationships with the Hurst exponent and the spectral index (as shown in Table S3). This relationship enables DFA to effectively indicate the presence of $1/f$ noise.

For revision, we have supplemented the rationale for method selection in the Temporal scaling analysis section of the Methods:

“We chose this method because of its broad applications, its alignment with our research objective of analyzing the scaling effects of temporal dynamics, and its ability to succinctly characterize the dynamics and long-term correlations of time-varying processes in a straightforward and easily interpretable manner, without requiring complex transformations.”

Reviewer #3 (Remarks on code availability):

Comment 7: Since the authors didn't provide the original data, I cannot reproduce the results. However, based on the readme file information, sample data and the Matlab code, I think the results can be replicated to other studies by following the methods.

Response:

We sincerely thank the reviewer for the positive feedback regarding code availability. In this revision, we have further improved the readability and operability of the code to

facilitate interpretation of the execution and computation process.

a) Reorganized code structure:

We reorganized the code by splitting the source code into a series of modular functions with explicit functionalities. These functions can now be executed by calling them through a central main function. This restructuring makes the code more organized, easier to interpret, and simpler to use.

b) Added interactive parameter settings:

We incorporated interactive prompts into the code, allowing users to set parameters (e.g., file names or key estimation parameters) during execution without needing to modify the source code directly. This enhancement ensures that users can easily repeat operations and use the code without requiring prior knowledge of the code structure or syntax.

c) Provided detailed code comments:

We added detailed comments throughout the code, explaining each function and key statements. These comments aim to make it easier for others to understand, compile, and modify the code for their own studies.

d) Uploaded additional data preprocessing codes:

We have also included data preprocessing codes for the Milan and Greater Boston datasets in the repository, providing further support for replicating the methods.

For more details, please refer to the updated code available in the GitHub/CodeOcean repository.

References

- Arcaute, E. 2020. Hierarchies defined through human mobility. *Nature*, 587, 372–373.
- Bak, P., Chen, K., and Creutz, M. 1989. Self-organized criticality in the ‘Game of Life’. *Nature*, 342(6251), 780-782.
- Barlacchi, G., De Nadai, M., Larcher, R., Casella, A., Chitic, C., Torrìsi, G., Antonelli, F., Vespignani, A., Pentland, A., and Lepri, B. 2015. A multi-source dataset of urban life in the city of Milan and the Province of Trentino. *Scientific Data*, 2(1), 150055.

- Bassingthwaighte, J. B. 1988. Physiological heterogeneity: fractals link determinism and randomness in structures and functions. *Physiology*, 3(1), 5-10.
- Batista e Silva, F., Freire, S., Schiavina, M., Rosina, K., Marín-Herrera, M. A., Ziemba, L., Craglia, M., Koomen, E., and Lavallo, C. 2020. Uncovering temporal changes in Europe's population density patterns using a data fusion approach. *Nature Communications*, 11(1), 4631.
- Batty, M., and Longley, P. A. 1994. *Fractal cities: a geometry of form and function*. London: Academic.
- Batty, M., and Xie, Y. 1999. Self-organized Criticality and Urban Development. *Discrete Dynamics in Nature and Society*, 1999(2-3), 109–124.
- Cannon, M. J., Percival, D. B., Caccia, D. C., Raymond, G. M., and Bassingthwaighte, J. B. 1997. Evaluating scaled windowed variance methods for estimating the Hurst coefficient of time series. *Physica A: Statistical Mechanics and its Applications*, 241(3-4), 606-626.
- Chen, Y. 2010. Exploring the fractal parameters of urban growth and form with wave-spectrum analysis. *Discrete Dynamics in Nature and Society*, 2010(1), 974917.
- Chen, Y., and Wang, J. 2013. Multifractal characterization of urban form and growth: the case of Beijing. *Environment and Planning B: Planning and Design*, 40(5), 884-904.
- Chen, Y., and Zhou, Y. 2008. Scaling laws and indications of self-organized criticality in urban systems. *Chaos, Solitons & Fractals*, 35(1), 85–98.
- DePetrillo, P. B., and Ruttimann, U. E. 1999. Determining the Hurst exponent of fractal time series and its application to electrocardiographic analysis. *Computers in Biology and Medicine*, 29(6), 393-406.
- Dong, L., Duarte, F., Duranton, G., Santi, P., Barthelemy, M., Batty, M., Bettencourt, L., Goodchild, M., Hack, G., Liu, Y., Pumain, D., Shi, J., Verbavatz, V., West, G. B., Yeh, Y., and Ratti, C. 2024. Defining a city: delineating urban areas using cell-phone data. *Nature Cities*, <https://doi.org/10.1038/s44284-023-00019-z>.
- Eichner, J. F., Koscielny-Bunde, E., Bunde, A., Havlin, S., and Schellnhuber, H. J. 2003. Power-law persistence and trends in the atmosphere: A detailed study of long temperature records. *Physical Review E*, 68(4), 046133.
- Eke, A., Herman, P., Kocsis, L., and Kozak, L. R. 2002. Fractal characterization of complexity in temporal physiological signals. *Physiological Measurement*, 23(1), R1.
- Frankhauser, P. 1994. *La fractalité des structures urbaines* [The fractality of urban structures]. PhD diss., Université de Paris 1.

- Ge, E., and Leung, Y. 2013. Detection of crossover time scales in multifractal detrended fluctuation analysis. *Journal of Geographical Systems*, 15, 115-147.
- Goldberger, A. L., Amaral, L. A. N., Hausdorff, J. M., Ivanov, P. C., Peng, C.-K., and Stanley, H. E. 2002. Fractal dynamics in physiology: Alterations with disease and aging. *Proceedings of the National Academy of Sciences*, 99(Suppl 1), 2466–2472.
- Höll, M., and Kantz, H. 2015. The relationship between the detrended fluctuation analysis and the autocorrelation function of a signal. *The European Physical Journal. B, Condensed Matter Physics*, 88(12), 1–7.
- Hurst, H. E. 1951. Long-term storage capacity of reservoirs. *Transactions of the American Society of Civil Engineers*, 116, 770-808.
- Jiang, S., Alves, A., Rodrigues, F., Ferreira Jr, J., and Pereira, F. C. 2015. Mining point-of-interest data from social networks for urban land use classification and disaggregation. *Computers, Environment and Urban Systems*, 53, 36-46.
- Jiao, L. 2015. Urban land density function: A new method to characterize urban expansion. *Landscape and Urban Planning*, 139, 26-39.
- Kantelhardt, J. W. 2009. Fractal and multifractal time series. In *Encyclopedia of Complexity and Applied Systems Science*, ed. Meyers, R. A., 3754–3778. New York: Springer.
- Kantelhardt, J. W., Koscielny-Bunde, E., Rego, H. H., Havlin, S., and Bunde, A. 2001. Detecting long-range correlations with detrended fluctuation analysis. *Physica A: Statistical Mechanics and its Applications*, 295(3-4), 441-454.
- Kantelhardt, J. W., Rybski, D., Zschiegner, S. A., Braun, P., Koscielny-Bunde, E., Livina, V., Havlin, S., and Bunde, A. 2003. Multifractality of river runoff and precipitation: comparison of fluctuation analysis and wavelet methods. *Physica A*, 330(1), 240–245.
- Mandelbrot, B. B., and Van Ness, J. W. 1968. Fractional Brownian motions, fractional noises and applications. *SIAM review*, 10(4), 422-437.
- Murcio, R., Masucci, A. P., Arcaute, E., and Batty, M. 2015. Multifractal to monofractal evolution of the London street network. *Physical Review E*, 92(6), 062130.
- Pappalardo, L., Manley, E., Sekara, V., and Alessandretti, L. 2023. Future directions in human mobility science. *Nature Computational Science*, 3(7), 588-600.
- Quinn, P. 2013. Road density as a proxy for population density in regional-scale risk modeling. *Natural Hazards*, 65, 1227-1248.
- Rozenfeld, H. D., Rybski, D., Andrade Jr, J. S., Batty, M., Stanley, H. E., and Makse, H. A. 2008. Laws of population growth. *Proceedings of the National Academy of Sciences*, 105(48), 18702-18707.

- Schläpfer, M., Dong, L., O’Keeffe, K., Santi, P., Szell, M., Salat, H., Anklesaria, S., Vazifeh, M., Ratti, C., and West, G. B. 2021. The universal visitation law of human mobility. *Nature*, 593(7860), 522–527.
- Simini, F., González, M. C., Maritan, A., and Barabási, A.-L. 2012. A universal model for mobility and migration patterns. *Nature*, 484(7392), 96–100.
- Simonsen, I., Hansen, A., and Nes, O. M. 1998. Determination of the Hurst exponent by use of wavelet transforms. *Physical Review E*, 58(3), 2779.
- Tannier, C., Thomas, I., Vuidel, G., and Frankhauser, P. 2011. A fractal approach to identifying urban boundaries. *Geographical Analysis*, 43(2), 211-227.
- Weber, R. O., and Talkner, P. 2001. Spectra and correlations of climate data from days to decades. *Journal of Geophysical Research*, 106(D17), 20131–20144.
- Zhou, Y., and Leung, Y. 2010. Multifractal temporally weighted detrended fluctuation analysis and its application in the analysis of scaling behavior in temperature series. *Journal of Statistical Mechanics: Theory and Experiment*, 2010(06), P06021.
- Zipf, G. K. 1942. The unity of nature, least-action, and natural social science. *Sociometry*, 5(1), 48–62.

RESPONSE TO REVIEWER COMMENTS

Reviewer #2

Remarks to the Author:

The authors took very seriously the many requests and questions I made in my original review, providing a detailed rebuttal that dealt systematically with each point.

Many issues I arose were due to a misunderstanding of the text, either due to text ambiguity (in which case the authors amended it by adding explicit remarks or fixing the terminology) or to a plain mistake on my side (in which case the authors explained it in clear terms).

The clarity issues in mathematical formulations and derivations have been fixed, mainly by either adding details to derivations or commenting them. The discussion seems now to me reasonably fluent and accessible.

In response to my issues on reproducibility of results, the authors also provided (in rebuttal letter) results of additional experiments on open data, to show their strong limitations but also to prove that they still fit the overall results obtained in the paper.

Details on the method and experiments were also added to the text.

Overall, I believe the authors did a great job in answering to my concerns, providing integrations where needed, and explaining/defending their work in a convincing way when all the answers were already there in the paper.

Response: We sincerely thank the reviewer for the valuable comments and for accepting our revisions.

Only two small points were left behind:

1. Comment 3, about the effects of aggregating rings of increasing radius, was slightly misinterpreted: while the values of F are correctly weighted considering the overall area covered, larger rings will simply aggregate more data. The question is mainly a vague doubt: could this have any effect on statistic aggregations, like reducing their variance or similar?

Response: We appreciate the reviewer for raising the issue regarding data statistics. As noted, theoretically, when the radius of a concentric ring increases, its area increases as well, which may incorporate more data into the calculation and, to some extent, affect the estimation of statistical measures. To minimize this potential impact, we have employed relatively narrow rings (based on the data’s spatial resolution) for measurement and normalized the area within the rings. This approach—based on concentric rings—aligns more closely with our definition of the variable $F(s,r)$ (stated in Result 1) and can theoretically yield approximately unbiased estimates provided the data resolution is sufficiently fine. As explained in our previous response, this method is a mature and commonly used means of measuring human mobility (Schläpfer et al. 2021), population density (Chen 2010), and urban land density (Jiao 2015). Furthermore, when using the concentric ring structure to measure urban phenomena, urban form also affects data aggregation. Specifically, even though the area of a ring increases with its radius, the available data may decrease due to the reduction in urbanized land and human settlement moving from the city center toward the periphery (please see Supplementary Fig. S1–6). In our study, the urban forms (Supplementary Fig. S1–6) of the six cases are distinct, leading to data aggregation in diverse ways (Fig. R1), but their scaling behaviors of population fluctuations were similarly modeled. This demonstrates that the method of data aggregation in our case does not necessarily affect the spatiotemporal scaling laws we revealed.

Fig. R1 The distance from the main city center and the effective area covered by data within the corresponding circular ring. The effective area represents the degree of data aggregation within rings at different distances.

References

- Chen, Y. 2010. Exploring the fractal parameters of urban growth and form with wave-spectrum analysis. *Discrete Dynamics in Nature and Society*, 2010(1), 974917.
- Jiao, L. 2015. Urban land density function: A new method to characterize urban expansion. *Landscape and Urban Planning*, 139, 26-39.
- Schläpfer, M., Dong, L., O’Keeffe, K., Santi, P., Szell, M., Salat, H., Anklesaria, S., Vazifeh, M., Ratti, C., and West, G. B. 2021. The universal visitation law of human mobility. *Nature*, 593(7860), 522–527.

2. Comment 4, part 2: I really expressed myself in the wrong way, apologies for that. My real aim was this: after stating (correctly) that (2) is a power law, we kind of forget the presence of a constant term containing "s", which might potentially be relevant in other derivations in the paper. Thus, I wanted to suggest to doublecheck that nowhere that really happened.

Response: We greatly appreciate the reviewer's careful attention to the form and interpretation of the spatiotemporal scaling law proposed in our study. Indeed, as you mentioned, when considering Eq. (2) as a power law, its constant term contains a temporal scale factor s (please refer to Eq. (13) where $C = \alpha_1 \ln s + Const$ is defined as described in Methods). However, we believe that this does not conflict with our interpretation of the scaling law in the main text. Below is our detailed explanation.

As stated in our previous response to your comment, Eq. (12) is the basic and original function form of Eqs. (1) and (2). If both variables (i.e., the spatial scale factor r and the temporal scale factor s) change simultaneously, the relationship cannot be interpreted as a power law. However, when s is not regarded as a variable but as a known constant value (as we stated that “Eq. (2) characterizes the scaling law of population dynamics in the spatial domain, indicating that **within a certain temporal interval s** , the population fluctuations $F(s, r)$ decrease in proportion to the increase of spatial distance r from the closest city center.” in Result 1), Eq. (13), which is achieved by reorganizing Eq. (12), can be transformed into Eq. (2), and at this point, both the constant and the power-exponent terms are numerically determined. For a simple example, let us take one hour and two hours as time scales (i.e., $s = 1$ hour and $s = 2$ hours, respectively) to measure the spatial scaling behavior of urban population fluctuations. Although this will result in a distinction in parameter values (i.e., different constant terms and scaling exponents that can be

estimated based on Eq. (13)), the function form remains a power law as expressed in Eq. (2).

Reviewer #2 (Remarks on code availability):

The code provided covers now all phases of the experiments discussed in the paper. Answering to the requests of reviewers, the code seems now complete, better structured and also (acceptably) commented. Some data sources used in the paper are not open, so nothing can be done to make experiment completely reproducible, but the code itself is there, and in a usable form to be applied/adapted to similar experiments.

Response: We thank the reviewer for reviewing our revised code and providing positive feedback.

Reviewer #3

Remarks to the Author:

The authors have done a great job to address all my comments and concerns. The paper has been significantly improved regarding its contribution, the clarity of research and equations to derive the spatiotemporal scaling law. It is a solid research to advance our understanding of the scaling law of urban human mobility not only in spatial scale but also in temporal scale.

Response: We cordially thank the reviewer for the constructive comments and the positive appraisal of our work and manuscript.

Reviewer #3 (Remarks on code availability):

The reorganized code is clear and easy to follow.

Response: We thank the reviewer for reviewing our code again and providing positive feedback.